# CultureVLM: Characterizing and Improving Cultural Understanding of Vision-Language Models

## Abstract

Vision-language models (VLMs) have advanced human-AI interaction but struggle with cultural understanding, often misinterpreting symbols, gestures, and artifacts due to biases in predominantly Western-centric training data. In this paper, we introduce **CultureVLM** for improved global cultural perception and understanding, with cross-cultural and cross-regional generalization while preserving general visual understanding and reasoning capabilities. To characterize and improve VLMs' multicultural understanding, we develop **CultureVerse**, a scalable multimodal cultural data collection and construction framework, which ultimately leads to a large-scale corpus covering 188 countries/regions, 15 cultural topics, and 3 question types. CultureVLM and CultureVerse effectively mitigate regional disparities, enabling cultural evaluation and enhancement to extend to low-resource regions. Further analysis and case studies also show that VLMs tend to produce fine-grained hallucinations in cultural understanding. We hope that this work could lay the foundation for more equitable and culturally aware multimodal AI systems.

## 1 Introduction

Vision-language models (VLMs) have achieved great performance in various tasks, such as visual question answering and captioning (OpenAI, 2024b; Hurst et al., 2024; Team et al., 2023; Anthropic, 2024; Wang et al., 2024; Liu et al., 2024b). Meanwhile, one of the most vital aspects of human experience, cultural understanding, which encompasses language, cultural values, social norms, culinary practices, and artistic expressions–remains a challenging area for these models (Winata et al., 2024; Adilazuarda et al., 2024).

**Challenges.** Cultural understanding is essential for AI systems intended for global deployment, as it enables them to interact appropriately and sensitively with users of diverse cultural, ethnic, and social backgrounds. However, current VLMs often struggle to grasp the deeper cultural meanings embedded in symbols and artifacts. For instance, a VLM may identify an eagle as merely a bird, overlooking its symbolic significance as a national emblem representing the spirit and identity of the United States. Similarly, the lotus flower is not only a plant, but a profound symbol of purity and spiritual enlightenment in Indian culture. Gestures present an even more complex challenge: the "OK" hand gesture, which conveys a positive meaning in North American countries, is interpreted as offensive in countries such as Brazil and Turkey (Medhat, 2015). Misinterpretations of culturally significant symbols can lead to misunderstandings and even cause offense.

These challenges partially stem from inherent biases and limitations in VLMs' training data: *1) Skewed Domain Coverage.* Pre-training images and texts predominantly feature *generic* daily scenes or natural settings, often lacking coverage of *culturally specific* artifacts, traditions, beliefs, and historical sites. Models may fail to interpret culturally significant symbols, particularly those from underrepresented regions. *2) English-centric Data and Western Bias.* The texts for pre-training VLMs is primarily sourced from English content (Naous et al., 2023a; Jin et al., 2024a), which predominantly represents high-resource cultures and introduces a Western bias, thereby limiting the models' understanding of diverse cultures (Young, 2014; Deng et al., 2024; Chiu et al., 2024).

Addressing these challenges is crucial for developing culturally aware AI systems that can engage effectively and respectfully with global users. Although there have been efforts to build culturally

Figure 1: Our pipeline to collect CultureVerse and build CultureVLM.

aware large language models (LLMs) through data collection (Shi et al., 2024; Li et al., 2024b; Chiu et al., 2024), improving VLMs for cultural understanding remains in its infancy. Existing multimodal cultural corpora (Romero et al., 2024; Nayak et al., 2024) usually rely on small-scale human annotation for data curation, and such unsystematic collection often lacks sufficient regional and national representation, failing to capture deep cultural relevance. From the *modeling* perspective, they cannot provide large-scale training to enhance models' cultural perception. From the *benchmarking* perspective, they tend to overlook low-resource and non-representative regions, resulting in incomplete evaluation and limiting the ability to analyze global trends from a broader perspective.

**This Work.** We take the first step toward enhancing VLMs' cultural understanding by introducing CultureVLM, a series of globally aware VLMs with targeted improvements across more than 188 countries and regions. To support this, we develop a scalable framework for multimodal cultural data collection and construction (Figure 1), resulting in a large-scale and high-quality multimodal cultural resource, CultureVerse. Our flexible pipeline encompasses dataset collection, quality verification, and filtering processes, and can easily incorporate additional languages and cultures, ensuring both high quality and inclusiveness. To enable a more comprehensive evaluation of VLMs, we also create a non-overlapping test set, carefully curated and verified by human experts, serving as a benchmark to assess VLMs' performance in low-resource regions and to conduct global analysis from a macro perspective. Our work lays a solid foundation for developing more equitable AI systems that meet the needs of developing nations, ethnic minorities, and underrepresented cultural groups.

**Key Findings.** We conduct extensive exploration and analysis of CultureVLM and CultureVerse from multiple perspectives. The key findings are as follows.

- **Disparity in Cultural Understanding**: *All* VLMs show highly consistent regional disparities in cultural understanding, with the highest cultural understanding for the Americas, followed by Europe and Oceania, and the weakest understanding for Asia and Africa.
- **Training for Enhanced Cultural Perception**: Fine-tuning effectively enhances the cultural perception of VLMs, narrowing the gaps in cultural understanding across different regions and categories without significantly compromising the model's general capabilities.
- **Model and Data Scale Enhance Cultural Understanding**: Cultural understanding is generally positively correlated with model size, though not absolutely, as demonstrated by the Llama 3.2-11B model achieving performance comparable to that of Qwen 2-72B. Regarding fine-tuning, larger training datasets lead to more significant improvements, but gains slow as data grows.
- **Generalization across Cultures, Concepts, Continent, and Datasets**: Due to the inherent correlations between cultures of different regions and types, fine-tuning for cultural understanding exhibits reasonable generalization across different cultures, concepts, continents, and even datasets, showing great potential to improve cultural understanding via generalization research.

**Contributions.** Our contributions are as follows:

- **Large-scale Dataset.** We present CultureVerse, a massive scale benchmark consisting of $19,682$ cultural concepts and $228,053$ samples, covering 3 tasks, 188 countries/regions, and 15 cultural topics. Over two-thirds of these countries/regions contribute more than 30 cultural concepts each. The testset includes $11,085$ widely recognized concepts and corresponding $31,382$ samples.
- **Comprehensive Evaluation.** We evaluate a variety of cultural concepts across 16 open-source and proprietary models of varying scales, and provide analyses from the perspective of different tasks.
- **Improvement of Multimodal Cultural Understanding.** We present CultureVLM, which includes a flexible and cost-effective data collection and construction process and a series of VLMs fine-tuned on our dataset. Experimental results show that CultureVLMs enhance cultural understanding while maintaining general capabilities and exhibiting a degree of generalization abilities.

## 2 RELATED WORK

**Cultural Bias in LLMs and VLMs.** Recent research has increasingly focused on cultural biases present in LLMs. Johnson et al. (2022) investigated conflicts between model outputs and input values and found that GPT-3's responses aligned more closely with dominant U.S. cultural norms. Similarly, Naous et al. (2023b) observed a bias toward Western cultural perspectives in models processing Arabic text. The Cultural Alignment Test, based on Hofstede's cultural dimensions framework (Geert Hofstede, 2010), was used to evaluate the cultural alignment of models like ChatGPT and Bard across various regions, showing that GPT-4 exhibited the strongest alignment with U.S. values (Masoud et al., 2023). Additionally, Cao et al. (2023) found that, while ChatGPT was well-aligned with American cultural values, it struggled to represent other cultures accurately, especially when responding to English prompts. Liu et al. (2023) further reported that multilingual LLMs showed limited proficiency in reasoning with proverbs and revealed a "culture gap" in translation.

**Data and Models for Cultural Understanding.** Most research adopt public datasets. (Wang et al., 2023) introduced a benchmark based on the World Values Survey (WVS) (Survey, 2022) and the Political Culture and Trust dataset (Mudde, 2016). Subsequent works include the Cultural Alignment Test (Masoud et al., 2023), NORMSAGE (Fung et al., 2022), WorldValueBench (Zhao et al., 2024), and NORMAD (Rao et al., 2024), each drawing on various existing datasets. Other sources include CultureAtlas (Fung et al., 2024) and MAPS (Liu et al., 2023), which collected data from Wikimedia, while Candle (Nguyen et al., 2023a) and CultureBank (Shi et al., 2024) gathered data from social media including TikTok and Reddit. In contrast, there is a growing trend toward data augmentation such as Li et al. (2024b;c). Recent work built culture-specific LLMs on large-scale datasets for alignment (Pires et al., 2023; Chan et al., 2023; Nguyen et al., 2023b; Pipatanakul et al., 2023; Abbasi et al., 2023; Lin & Chen, 2023). Instead of relying on manual data collection, Li et al. (2024b;c) proposed cost-efficient approaches to fine-tuning cultural-specific LLMs with data augmentation.

Unlike LLMs, VLMs face a more severe situation in obtaining sufficient cultural training data. Such research is still preliminary, with most efforts in *manual* data collection (Liu et al., 2021; Romero et al., 2024; Nayak et al., 2024; Bhatia et al., 2024; Vayani et al., 2024; Baek et al., 2024; Thapliyal et al., 2022). MaRVL (Liu et al., 2021) built an ImageNet-style hierarchy that represents a wider range of languages and cultures. CVQA (Romero et al., 2024) proposed a culturally diverse multilingual VQA benchmark to

Table 1: Comparison of various cultural datasets with features. 'Multi.' indicates whether the dataset provides multi-faceted questions rather than just one single type.

| Dataset | Country | Concept | Image | Question | Multi. | Trainset |
|---|---|---|---|---|---|---|
| MaRVL (Liu et al., 2021) | 5 | 454 | 4,914 | 5,670 | ✗ | ✗ |
| CVQA (Romero et al., 2024) | 28 | - | 4,560 | 9,044 | ✗ | ✗ |
| CulturalVQA (Nayak et al., 2024) | 11 | - | 2,328 | 2,328 | ✗ | ✗ |
| GlobalRG (Bhatia et al., 2024) | 50/15 | - | 3,591 | - | ✓ | ✗ |
| ALM-Bench (Vayani et al., 2024) | 73 | - | 2,929 | 22,763 | ✓ | ✗ |
| CultureVerse | 188 | 11,085 | 11,085 | 31,382 | ✓ | ✓ |
| w/ Train Set | 188 | 19,682 | 74,959 | 196,673 | | |

encompass a wide variety of lingual and cultural contexts, engaging native speakers and cultural experts for data collection. CulturalVQA (Nayak et al., 2024) developed a visual question-answering benchmark focused on evaluating VLMs' understanding of culturally diverse, geographically specific content. GlobalRG (Bhatia et al., 2024) presents two challenging tasks: retrieval across cultural universals and culturally specific visual grounding. ALM-Bench (Vayani et al., 2024) expanded the languages of the test data to 100 through large-scale machine translation.

However, these datasets are often built by randomly sampling images rather than ***concept-driven***, typically limited in scale and coverage, with insufficient regional and national representation. More critically, at the model level, there has been no effort to develop culturally aware VLMs, posing a major barrier to AI equity for underrepresented cultures. By systematically constructing datasets and models through *tangible concepts*, we achieve both *reduced manual effort* and ensured *scalability*. Table 1 shows the key difference between our benchmark and existing multimodal cultural datasets.

## 3 CULTUREVERSE FOR SCALABLE CULTURAL CORPUS CONSTRUCTION

Collecting reliable culture datasets presents two key challenges: *Diversity* and *Scalability*. Achieving comprehensive coverage is especially difficult for *culturally diverse* topics, particularly for underrepresented groups in the Global South. The construction and annotation of such datasets would typically require substantial local residents from various countries and ethnic groups, resulting in poor scalability and high costs. Existing cultural benchmarks for VLMs usually lack adequate representation of diverse regions and communities, and often reflect a bias towards dominant cultures (Liu et al., 2021; Romero et al., 2024; Oh et al., 2024).

To overcome these limitations, we introduce a scalable data collection pipeline that integrates automated web crawling for *scalability* and *diversity* with expert human annotation for *reliability*. Our pipeline (Figure 1) consists of three stages: tangible cultural concept collection, question-answer generation, and quality assurance. This hybrid approach ensures that our dataset captures a wide spectrum of cultural contexts while maintaining high standards of data quality and relevance.

## 3.1 TANGIBLE CULTURAL CONCEPT COLLECTION

To construct a comprehensive set of cultural concepts, a common approach is to employ a bottom-up strategy to retrieve specialized knowledge from open web documents. For example, Fung et al. (2024) begin with an initial set of cultural topics (e.g., education and holiday traditions), collect relevant Wikipedia documents, and expand their scope through linked connections. However, many resulting documents primarily describe general, abstract, or high-level concepts, such as `Renaissance Art` or `Mediterranean cuisine`, which often lack specific, unambiguous visual representations.

**Concept Construction.** To overcome this issue, we adopt a top-down approach, starting with 15 predefined cultural categories of tangible cultural concepts such as food, festivals, landmarks, and performing arts (Figure 2 and Table 2). These categories are chosen to capture culturally distinctive and visually recognizable elements suitable for image retrieval and analysis. We then use GPT-4o[1] to process all relevant Wikipedia documents, extracting conceptual entities that align with the 15 predefined categories. To ensure the quality and specificity of the extracted entities, we implement a 3-step filtering process on the extracted conceptual entities: 1) *Entity Consolidation.* We unified duplicate entities, merging those that are identical or differ solely by case, and eliminated entities with formatting issues or irregularities; 2) *Frequency-based Thresholding.* We retained only entities that appeared at least twice across the documents from a country, ensuring these concepts are well-recognized within their cultural context and preventing formatting errors; 3) *Entity Refinement.* We filtered out overly abstract or generic entities such as `Imperial Cuisine` and those lacking distinct regional specificity such as `Steak` using additional judgment by GPT-4o. Through this refined process, we curated a collection of over $19,682$ cultural concepts from $188$ countries, as shown in Table 10. Our pipeline ensures that the selected concepts are diverse and recognizable, relevant for evaluating VLMs in understanding global cultural diversity.

**Image Retrieval.** Using these concepts and their corresponding countries, we use the Google image retrieval tool to scrape images from Google Images for each cultural concept, obtaining 5 images for each concept. After filtering for quality, the number of images per concept varies and does not always reach five. During the development of CultureVLM, we found that no existing benchmark could adequately evaluate the performance of our intermediate models, as none provided coverage comparable to our corpus. Therefore, we reserved the first image for a human-curated test set and allocated the remaining images to the training set. Images larger than 10MB were compressed to ensure compatibility with typical input requirements of VLMs.

## 3.2 QUESTION-ANSWER GENERATION

We designed three levels of VQA tasks to assess and improve the multicultural knowledge of VLMs:

**Image Recognition Questions** evaluate models' ability to identify cultural concepts in images. Accurate identification of such concepts is fundamental to retrieving relevant cultural knowledge. Given an image and a cultural concept, models answer questions like "What dish is in the image?"

**Cultural Knowledge Questions** further evaluate model's deeper understanding of the cultural background associated with the concepts. For each concept, we generate comprehensive descriptions, including aspects like location, characteristics, history, and cultural significance. Then, we instruct GPT-4o to formulate a question based on the introduction and image to probe this cultural knowledge without directly naming the concept. These questions require the model to identify cultural concepts and apply various levels of reasoning, drawing on relevant cultural knowledge for accurate answers.

**Scene Understanding Questions** are designed to assess the model's ability to interpret, interact, and respond within culturally specific contexts, rather than simply recalling factual information as in the previous two categories. We curate scenarios with cultural elements or characteristics depicted in the

---

[1]GPT-4o has strong performance for *text*-based cultural understanding (Li et al., 2024c).

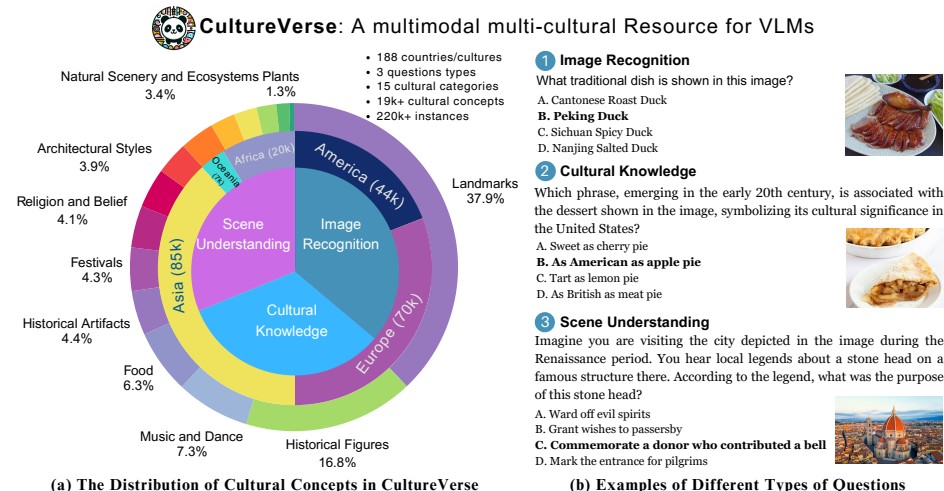

(a) The Distribution of Cultural Concepts in CultureVerse     (b) Examples of Different Types of Questions

Figure 2: Overview of CultureVerse. There are over 220k instances and 19k cultural concepts for training and evaluation, composed of 3 different types of questions from 188 countries.

images, providing context cues that challenge the VLMs to make contextually appropriate choices. Using the detailed introductions of the previous steps, we prompt GPT-4o[2] to generate scenario-based understanding questions. These questions require the model to not only recognize cultural concepts but also apply contextual reasoning based on the associated cultural knowledge.

## 3.3 QUALITY ASSURANCE

We pursue an *optimal integration* of LLM assistance and human expert annotation to ensure both accuracy and scalability in dataset construction. To ensure the quality of CultureVerse, every cultural concept—together with its paired images and QA triplets underwent a rigorous quality-control pipeline in which erroneous samples were removed or corrected. Concretely, we applied multiple LLM-based filtering rounds to the full corpus and engaged human experts to refine the test set (details are in Appendix D), focusing on three dimensions:

- **Image-Concept Alignment.** We assessed whether each cultural concept accurately represents the culture of its respective country or region and is either unique to or widely recognized within that. We started with frequency analysis and leverage GPT-4o for preliminary screening, effectively filtering out less desirable data and significantly reducing the manual review workload.
- **Image Quality Check.** We checked the image quality and ensured that the cultural concept is accurately presented in the image, filtering out non-matching or low-quality instances.
- **Question & Answer Validation.** We verified that all generated questions are reasonable, clear, logically sound, and have a single correct answer. Annotators refined the questions and answers by removing redundant information and resolving any ambiguities to maintain clarity and accuracy.

Following the quality assurance process, we utilized human annotations for the evaluation set of CultureVerse while applying the automated annotation pipeline to the larger training set. With over 90% of the evaluation set samples correctly annotated by the LLMs-based filtering (Appendix D.2), we conclude that the pipeline is highly effective. Any remaining erroneous or challenging samples that could not be refined were filtered out to maintain the dataset's high quality.

## 3.4 DISCUSSION ON SCALABILITY

Our approach is notably more scalable and comprehensive than existing methods which mostly rely on *manual* efforts to search for cultural concepts, retrieve images, and formulate questions, significantly increasing human efforts beyond quality check (Chiu et al., 2024; Romero et al., 2024; Jin et al., 2024b). This manual process typically results in limited or biased coverage due to the

---

[2]As for *cultural image* understanding, GPT-4o may perform worse, motivating the quality assurance in §3.3.

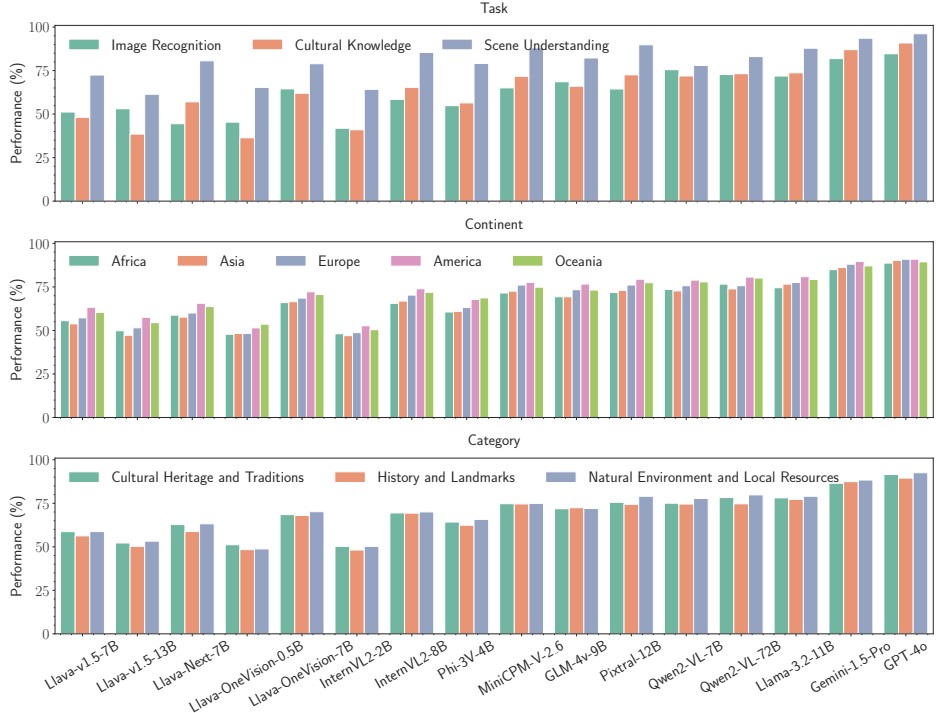

Figure 3: Accuracy of different models on three tasks (upper), five regions (middle), and three categories of concepts (lower).

limited scope of cultural concepts explored. For example, some datasets (Bhatia et al., 2024) cover only 15 countries with 40 cultural concepts for each country.

In contrast, our work is the first to advance the number of cultural concepts to the scale of tens of thousands, with a coverage of $\sim$ 200 countries/regions (Table 10). Additionally, our dataset construction process allows for further large-scale expansion, including the retrieval of more images and the synthesis of QAs, such as open-ended QAs, multiple-choice and reasoning QAs. Note that our work is *not* replacing human annotations; expert annotation is still valuable, and the combination of expert and automatic pipeline provides a promising solution for massive cultural data collection.

**Dataset Analysis.** Figure 2 (a) illustrates the distribution of three tasks (Section 3.2), 5 continents (188 countries, with North and South America combined into America) and 15 cultural topics in CultureVerse. Asia accounts for the largest scale of cultural concepts as the birthplace of multiple civilizations and religions, and non-Western cultures comprise nearly half of the total. Africa has relatively fewer cultural concepts, and systematically collecting data from public sources remains a significant challenge despite its large number of countries. Detailed counts of countries and concepts are provided in the Table 10. Despite the limited web data in non-representative regions, over 120 regions still yield more than 30 concepts each, offering a more comprehensive perspective. Detailed statistics of the concepts are provided in Table 1. Compared to recent multimodel culture datasets, CultureVerse is driven by *tangible, presentable cultural concepts*, achieving an order of magnitude increase in the number of regions, images, and questions. This expansion advances multimodal and multicultural research beyond a limited set of countries, moving toward truly global, inclusive cultural integration. Figure 2(b) shows examples of three questions from one concept, clearly demonstrating that different question types evaluate different abilities.

## 4 EXPERIMENTS WITH CULTUREVERSE

### 4.1 EXPERIMENTAL SETUP

**Data Split.** To ensure robust evaluation, we partition CultureVerse into training/test sets for all countries/regions, allowing us to assess transferability between regions. We select more common cultural concepts from the entire dataset for the test set, which underwent manual quality checks, while the training set includes all cultural concepts. We ensured that the images in training and test sets did not overlap to prevent data leakage. More details are in Appendix C.1.

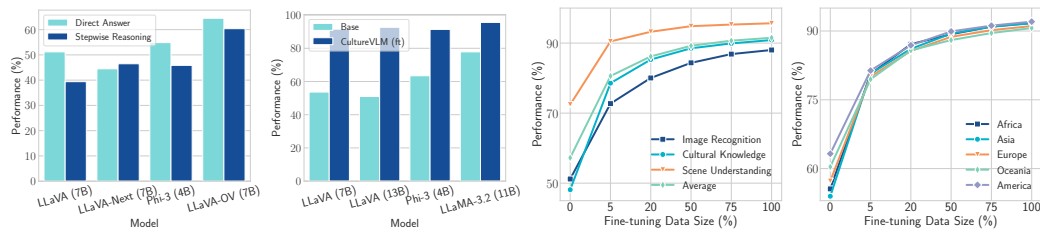

(a) Direct v.s. reasoning    (b) Fine-tuning results    (c) Cultural concepts    (d) Continents

Figure 4: Results and analysis of CultureVLM by fine-tuning on our CultureVerse. Performance (%) indicates the accuracy of VLMs in cultural understanding tasks: (a) compares direct output and stepwise reasoning performance on the image recognition task; (b) shows base models' performance before and after fine-tuning; (c) and (d) depict performance variations with different fine-tuning data sizes and across three tasks and five continents, respectively.

**Models and Hyperparameters.** We evaluate CultureVerse on 14 open-source and 2 proprietary VLMs. For multi-choice questions, we employ greedy search decoding for deterministic predictions. We use vllm (Kwon et al., 2023) and lmdeploy (Contributors, 2023) to speed up inference. We report accuracy following previous work (Liu et al., 2024b). More details are in Appendix C.1.

## 4.2 MAIN RESULTS

The main results are in Figure 3 with details in Appendix C.2. Our main findings are as follows.

**Task Characteristics: Cultural Awareness Outpaces Image and Detail Recognition.** From the task perspective, we observe that image recognition and cultural knowledge questions pose challenges comparable to VLMs. *Image recognition* tests VLMs' ability to identify culturally specific objects or concepts, which relies heavily on diverse and relevant image data (Pouget et al., 2024). For instance, recognizing traditional foods like *kimchi* from Korea, or regional attire such as a *sari* from India, requires the model to have encountered similar image-text pairs in its training data. In contrast, cultural knowledge questions assess the model's understanding of broader cultural elements based on text-based training. For example, asking about the significance of a festival like *Diwali* or the symbolism of a *red envelope* during Lunar New Year taps into the model's text-based memory, which tends to be richer due to the abundance of internet text data. Interestingly, in the *scene understanding* task, which integrates images with contextual background (e.g., a Japanese tea ceremony scene or a Brazilian Carnival parade), VLMs tend to generate culturally appropriate responses, avoiding inappropriate or culturally insensitive outputs. This can be attributed to the model's inherent multicultural awareness and its alignment with ethical and harm-reduction training.

**Regional Disparities: Better Performance in Western Cultures.** A dominant *regional disparity* is observed among all models: VLMs demonstrate the strongest cultural understanding of the Americas, followed by Europe and Oceania. This trend reflects the dominance of English data centered around the Global North, leading to a disproportionate focus on Western cultural content. North America's relatively homogenous cultural landscape, combined with fewer countries, contributes to better model performance. In contrast, Asia and Africa show significantly weaker results, likely due to the scarcity of digitized, English-language data and the high cultural diversity in these regions. For instance, Asia consists of many countries with distinct and complex cultural contexts, such as those from East Asia, South Asia, and Southeast Asia, both within and across nations. Although Asia has the most data in CultureVerse (see Figure 2), the models struggle to capture the intra-regional and inter-regional cultural nuances, resulting in suboptimal performance.

**Weak Understanding of History and Landmarks.** VLMs generally exhibit weaker recognition and understanding of cultural concepts related to history and landmarks. The primary reason is the relatively limited internet data available on historical figures and landmarks. Additionally, recognizing a landmark typically requires training data that includes images from multiple perspectives to form a comprehensive, three-dimensional understanding.

**Performance Variability from Model Level.** Proprietary models continue to outperform open-source counterparts, with GPT-4o achieving the best results. Although larger models tend to demonstrate better performance, size alone is not the determining factor. For example, the size variations of LLaVA-1.5 and Qwen2-VL show similar performance. Cultural knowledge often resides in model's

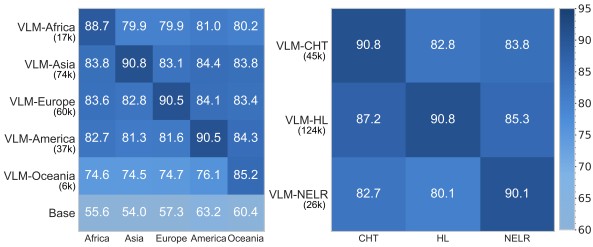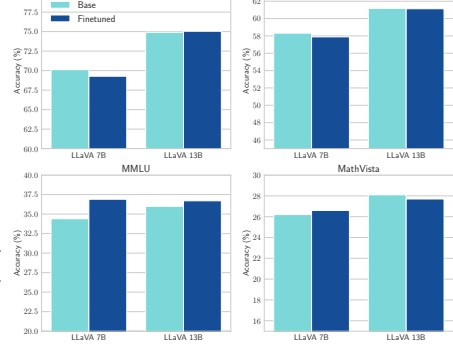

Figure 5: **Generalization and Robustness. Left**: Performance of CultureVLM (y-axis) evaluated across different continents (x-axis). CultureVLM achieves the highest performance for in-distribution settings, while demonstrating strong generalizability for out-of-domain settings. **Right**: CultureVLM fine-tuned on different CultureVerse categories (x-axis) and evaluated across others (y-axis). CHT denotes *Cultural Heritage and Traditions*; HL denotes *History and Landmarks*; and NELR denotes *Natural Environment and Local Resources*.

Figure 6: Performance of open-source and proprietary models on general VQA datasets. *Base* refers to the original model, while *Finetuned* refers to the model adapted using our CultureVerse.

memory, an aspect overlooked in VLMs development (Ananthram et al., 2024). Thus, smaller models (e.g. Phi-3-Vision) with comparable training data can already exhibit strong cultural understanding when using similar training data as larger models.

**Direct Answer v.s. Stepwise Reasoning.** For image recognition tasks, we compare two prompt methods: 1) the model directly identifies and outputs the cultural concept, and 2) first provides a detailed image description and then analyzes and compares the options to reach a final answer. As shown in Figure 4a, we find that stepwise reasoning does not improve cultural recognition and, in most cases, significantly *impairs* performance. Upon analyzing the answers, we observe that VLMs frequently exhibit hallucinations (Bai et al., 2024; Tonmoy et al., 2024; Agarwal et al., 2024) during step-by-step explanations, as shown in Figure 8. This poor robustness suggests that while VLMs may have encountered similar images during training and associated them with certain concepts, they may lack a deeper understanding of the details and components that make up those concepts.

### 4.3 TRAINING CULTUREVLM

We fine-tuned three models: LLaVA-1.5, Phi-3-Vision, and LLaMa-3.2-Vision, with the results presented in Figure 4b. Cultural knowledge, in contrast to reasoning tasks such as mathematics and coding, is relatively easier to enhance as a form of memory-based perception. Consequently, all models achieved consistent and substantial improvements, reaching performance levels comparable to closed-source models, with the most significant gains observed in Asia and Africa, thereby promoting inter-regional balance (Table 4). We also performed ablation studies to analyze the impact of fine-tuning data size and decoding temperature.

**Impact of Fine-tuning Data Size.** We vary the number of fine-tuning examples within $[5\%, 20\%, 50\%, 75\%, 100\%]$. Figure 4c and 4d show the model's performance decreases as the training data is reduced. However, the decline is minimal, indicating that even a small amount of training data can effectively enhance the model's multicultural awareness.

**Impact of Decoding Temperature.** We evaluate the performance of the original models and CultureVLMs under different temperature and decoding settings in Appendix C.3. It can be observed that VLMs perform better when the temperature is lower and decoding diversity is reduced. However, when the temperature reaches 1.0, there is a noticeable and expected decline in performance.

### 4.4 GENERALIZATION AND ROBUSTNESS

To evaluate the generalizability of VLMs in multicultural contexts, we partitioned the training data by continents (Americas, Asia, Europe, Africa, Oceania) and fine-tuned LLaVA-1.5-7B (Liu et al., 2024b) on each subset. Table 5 in the appendix illustrates the performance of each model trained on specific continental data and tested in all regions, and Figure 5 shows the aggregated results.

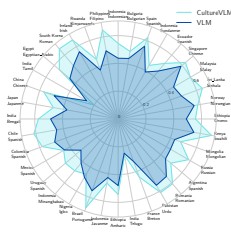

Figure 7: Results on CVQA ([Romero et al., 2024](#)).

Figure 8: Case study on the effect of fine-tuning and prompt variations.

**Intra- and Inter-continent Generalization.** The diagonal values represent cases where the model was trained and evaluated on data from the same continent, consistently yielding the highest scores. Notably, Asia achieved the best performance (90.8), followed closely by Europe and the Americas (90.5), indicating strong regional specialization. Furthermore, the off-diagonal values reveal models' ability to generalize across regions. Although cross-region scores are generally lower, the model still exhibits reasonable transferability. Fine-tuning on Oceania, however, resulted in the lowest average performance drop, from 85.2 to 75.0, suggesting a distinct data distribution for this region.

**Robustness across Concepts.** We grouped the 15 concepts into 3 main classes: *Cultural Heritage and Traditions* (CHT), *History and Landmarks* (HL), and *Natural Environment and Local Resources* (NELR) using GPT-4o. We then conducted training in each group and evaluated on all groups. The specific category mappings are in Appendix B. As shown in Figure 5 (right), models generally perform the best when trained and evaluated within the same category (in-distribution). The model fine-tuned on HL exhibits greater generalization, achieving an average performance of 87.8, compared to NELR (84.3). High off-diagonal scores, such as CultureVLM-HL's 87.2 when tested on CHT, indicate substantial cross-category knowledge transfer, particularly between culturally related classes.

**Cross-dataset Generalization.** We also evaluated the generalization ability of CultureVLM for cultural reasoning on other datasets. We tested LLaVA-1.5-7B on CVQA ([Romero et al., 2024](#)) and CulturalVQA ([Nayak et al., 2024](#)) before and after fine-tuning, as shown in in Figure 7 and Table 10. It is evident that our CultureVLM achieves improvements across most cultures and gets 22.56% improvement on average on CulturalVQA, which can be attributed to the comprehensive coverage of our dataset global nations and cultures, indicating the potential of CultureVLM for cultural research.

## 4.5 Catastrophic Forgetting and Case Study

Catastrophic forgetting ([Kirkpatrick et al., 2017](#)) happens when a model loses previously learned knowledge when trained on new information, especially when the new data diverges significantly from the pretraining data. This can be especially problematic in cultural knowledge acquisition, as it may cause the model to compromise essential commonsense knowledge in favor of culture-specific details. To assess this, we evaluate the models on standard VQA benchmarks, including ScienceQA ([Saikh et al., 2022](#)), TextVQA ([Singh et al., 2019](#)), MMLU ([Hendrycks et al., 2020](#)) and MathVista ([Lu et al., 2024](#)) to determine if the process of acquiring cultural knowledge inadvertently diminishes their grasp of general commonsense concepts. The results in Figure 6 reveal that our fine-tuned CultureVLM merely influences general VLM tasks, indicating the versatility of the solution.

Figure 8 shows the responses of LLaVA and CultureVLM. We incorporate extensive explanations into the training data, enriching CultureVLM with substantial knowledge that enhances its cultural recognition and understanding ability. More analysis and case studies are in Appendix E.

## 5 Conclusion

We constructed CultureVerse, a large-scale multimodal resource for VLM cultural understanding. Extensive evaluation shows significant performance disparities across regions and tasks, highlighting VLMs' cultural biases and weak performance in underrepresented regions. The fine-tuned CultureVLM improved cultural perception and cross-cultural generalization. Our findings underscore the importance of culturally diverse training data and provide actionable insights to improve VLMs.

We acknowledge several limitations. First, we use language as a proxy for cultural boundaries, primarily extracting knowledge and information from Wikipedia documents. However, Wikipedia itself carries potential initial biases, and our knowledge extraction pipeline may also introduce unintended biases. Moreover, the richness of culture extends beyond language and is better captured through multiple dimensions such as accents, social norms, rituals, and everyday practices. We adopt this simplification to pragmatically address the well-documented challenges in formally defining cultural contexts, following precedents in prior work (Li et al., 2024b; Appadurai, 1996; Myung et al., 2024). Our pipeline is flexible to incorporate additional languages and cultures. Second, current dataset lacks multilingual support. Most foundational models currently exhibit weak multilingual capabilities, so fine-tuning on multilingual cultural data is less effective than on English data (Li et al., 2024b). Therefore, the English data provided currently is also timely and valuable. Third, CultureVerse only contains multiple-choice questions. Exploring open-ended questions could also offer additional avenues for assessment.

## ETHICAL STATEMENT

For our corpus and models, we have implemented multiple safeguards, such as content filtering and screening, to prevent potential harm to individuals or groups. Given the social sensitivity of cultural understanding research, we wish to reiterate that all data and models are intended solely for research purposes, and will be governed through protocols and licenses.

## REPRODUCIBILITY STATEMENT

We have taken comprehensive steps to ensure the reproducibility of our results. Code and data will be made publicly available after careful review to support academic research within the community. Public benchmarks used in this work further support verifiability and consistency.

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

**Appendix**
# CultureVLM: Characterizing and Improving Cultural Understanding of Vision-Language Models

CONTENTS

## A  LLM USAGE STATEMENT

We utilized ChatGPT and Grammarly to polish the language of the manuscript. LLMs were employed solely for language refinement and editing purposes, such as improving readability and clarity. They played no role in shaping research ideas, methodology, or analysis. The authors assume full responsibility for all content.

## B  DETAILS OF THE CULTUREVERSE DATASET

Table 2 shows the definition of cultural concepts in our dataset. Table 10 shows the number of concepts in different countries from the evaluation set. In total, we have $11,085$ evaluation samples and $19,682$ training samples from 188 countries/regions.

| Overall Category | Category | Example Description |
|---|---|---|
| Cultural Heritage and Traditions | Festivals | Unique festival celebration scenes |
| | Traditional Clothing | Ethnic clothing, festival attire |
| | Handicrafts and Artifacts | Ethnic handicrafts, traditional handmade items |
| | Music and Dance | Traditional musical instruments, dance scenes |
| | Religion and Belief | Temples and churches, religious ceremonies |
| | Entertainment and Performing Arts | Theater performances, street performers |
| History and Landmarks | Famous Landmarks | Famous historical sites, buildings |
| | Historical Artifacts | Museum collections, ancient relics |
| | Historical Figures | Portraits of historical figures, statues |
| | Architectural Styles | Traditional architecture, modern landmark buildings |
| Natural Environment and Local Resources | Food | Local specialty dishes, traditional festival foods |
| | Plants | Unique flowers, crops in a certain area |
| | Animals | Unique wild animals, livestock in a certain area |
| | Natural Scenery and Ecosystems | Unique natural landscapes, ecological reserves |
| | Markets and Shopping Traditions | Local markets, specialty shops |

Table 2: Collected concepts, their overall categories, and descriptions in the dataset.

## C  EXPERIMENTAL DETAILS AND RESULTS

### C.1  EXPERIMENT SETUP

**Evaluation Models.** We conduct evaluations on the following models: (a) open-source models including LLaVA-1.5 (Liu et al., 2024a), LLaVA-1.6-Mistral-7B-Instruct (Liu et al., 2024b), LLaVA-OneVision (Li et al., 2024a), LLaMA-3.2-Vision (Meta, 2024), Qwen2-VL (Wang et al., 2024), InternVL-2 (Chen et al., 2023), Phi-3-Vision (Abdin et al., 2024), MiniCPM-Llama3-V-2.5 (Yao et al., 2024), GLM-4V (GLM et al., 2024), Pixtral 12B (Agrawal et al., 2024); (b) proprietary models such as GPT-4o (OpenAI, 2024a), Gemini-1.5-Pro (Team et al., 2023).

**Evaluation Setup.** For LLaVA-1.5, LLaMA-3.2-Vision and InternVL-2, we use lmdeploy Contributors (2023) for inference acceleration. For other models, we use vllm Kwon et al. (2023) for the acceleration of inference. For models before and after fine-tuning, we use the same acceleration toolkit to prevent potential impact. The number of questions differs across the three tasks. This is because generating questions for cultural knowledge and scenario reasoning is more complex, and in some cases, GPT-4o refused to provide answers, making it impossible to generate valid questions. For image recognition questions, we directly use the questions and options as prompts. For cultural knowledge and scene understanding questions, we employ stepwise reasoning prompts to facilitate the reasoning explanation. The prompts are available in the Appendix F. For all proprietary models, we utilize the default hyper-parameters, setting the temperature to 0 and the max tokens to 1,024. For all open-source models, $do\_sample$ is set to False, $max\_gen\_len$ is set to 512, and the temperature is set to 0.01.

**Training Setup.** We use the official train ing scripts of LLaVA[3], Phi[4], and LLaMA[5] for model training, largely adhering to the original hyperparameters, except for appropriately adjusting the batch size to accommodate the GPU memory capacity. For LLaVA-1.5, a learning rate of $2 \times 10^{-5}$ is used, with no weight decay applied (0.0). The learning rate followed a cosine schedule, gradually increasing during the initial phase with a warmup ratio of 0.03. For Phi-3, we use a learning rate of $4 \times 10^{-5}$ and a weight decay of 0.01. A linear learning rate scheduler is utilized, with 50 warmup steps to stabilize the early training stage. For LLaMA-3.2, fine-tuning is conducted using a learning rate of $1 \times 10^{-5}$ with no weight decay (0.0). A multiplicative learning rate decay is applied after each epoch, with a gamma value of 0.85. The batch sizes are set to 64, 16 and 32 respectively. All models are trained for one epoch on the training set and fully fine-tuned on 4×A100 80GB GPUs. For the training data, although we do not conduct large-scale human annotation, we synthesize the data using only concepts that passed either GPT-4o or human quality assurance, significantly improving the accuracy of the dataset. The prompts used for GPT quality check can be found in the Appendix F.

## C.2 DETAILED MAIN RESULTS

Detailed results on different tasks, continents, and cultural categories can be found in Table **??**.

## C.3 DETAILED FINE-TUNING RESULTS

The detailed results of the fine-tuned models are shown in Table 3. Detailed results for different temperature settings can be found in Figure 9. Detailed results on the generalization of the fine-tuned model in different regions and for different categories can be found in Table 5 and Table 6. Detailed results of the models before and after fine-tuning on the general VQA benchmark are shown in Table 7.

| Model | Task | | | Continent | | | | | Category | | |
|---|---|---|---|---|---|---|---|---|---|---|---|
| | Image Recognition | Cultural Knowledge | Scene Understanding | Africa | Asia | Europe | America | Oceania | CHT | HL | NELR |
| LLAVA-v1.5 7B | 88.03 | 90.87 | 95.66 | 91.68 | 91.61 | 91.02 | 91.91 | 90.61 | 92.58 | 90.98 | 91.70 |
| LLAVA-v1.5 13B | 89.72 | 92.20 | 96.09 | 92.68 | 92.37 | 92.59 | 93.05 | 92.73 | 93.17 | 92.40 | 92.60 |
| Phi-3-vision 4B | 87.53 | 90.84 | 95.77 | 90.80 | 91.56 | 91.41 | 90.91 | 91.16 | 92.83 | 90.50 | 92.35 |
| LLaMA-3.2-Vision 11B | 89.13 | 91.49 | 96.20 | 91.99 | 92.24 | 91.82 | 93.08 | 91.34 | 93.20 | 91.78 | 92.50 |

Table 3: Performance of fine-tuned models across three tasks, five continents, and three categories. CHT denotes *Cultural Heritage and Traditions*; HL denotes *History and Landmarks*; and NELR denotes *Natural Environment and Local Resources*.

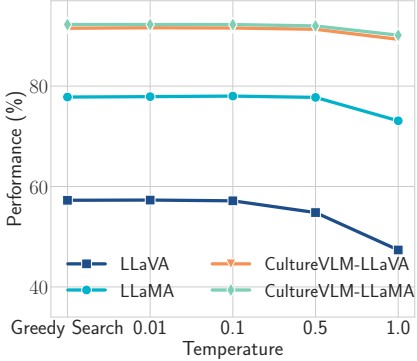

Figure 9: Impact of different decoding temperatures on performance

| Model | Score |
|---|---|
| LLaVa | 32.8 |
| CultureVLM-LLaVa-7B | 40.2 (+22.56%) |

Figure 10: CultureVLM (based on LLaVa-7B) achieved a 22.56% improvement on CultureVQA (Nayak et al., 2024). These results were obtained through an evaluation we requested from the CulturalVQA team.

---

[3] https://github.com/haotian-liu/LLaVA.
[4] https://github.com/microsoft/Phi-3CookBook.
[5] https://github.com/meta-llama/llama-recipes/blob/main/recipes/quickstart/finetuning/finetune_vision_model.md.

| Model (Improvement) | Africa | Asia | Europe | America | Oceania |
|---|---|---|---|---|---|
| LLaVA-7B | 36.06 | 37.83 | 33.78 | 28.64 | 30.20 |
| LLaVA-13B | 42.85 | 45.14 | 41.07 | 35.52 | 38.22 |
| Phi-3-vision-4B | 30.20 | 30.57 | 28.20 | 23.13 | 22.47 |
| LLaMA-3.2-Vision-11B | 17.44 | 15.62 | 14.25 | 12.18 | 12.06 |
| Average Improvement | **31.64** | **32.29** | 29.32 | 24.87 | 25.74 |

Table 4: Performance improvement across different cultural regions. CultureVLM demonstrates the most significant improvements in Africa and Asia, both exceeding +30 points, highlighting its enhanced effectiveness for regions traditionally underrepresented in cultural understanding.

| Task | Model | Total | Africa | Asia | Europe | America | Oceania |
|---|---|---|---|---|---|---|---|
| Image Recognition | LLaVA-v1.5-7B-Africa | 72.73 | 85.79 | 72.69 | 70.54 | 71.37 | 69.74 |
| | LLaVA-v1.5-7B-America | 75.84 | 75.35 | 73.24 | 72.52 | 86.00 | 76.05 |
| | LLaVA-v1.5-7B-Asia | 80.20 | 75.78 | 88.62 | 75.04 | 75.84 | 75.26 |
| | LLaVA-v1.5-7B-Europe | 79.34 | 77.61 | 75.64 | 86.89 | 75.62 | 76.05 |
| | LLaVA-v1.5-7B-Oceania | 64.51 | 64.05 | 63.14 | 63.60 | 65.28 | 83.95 |
| Cultural Knowledge | LLaVA-v1.5-7B-Africa | 78.54 | 86.77 | 75.82 | 78.28 | 80.38 | 79.42 |
| | LLaVA-v1.5-7B-America | 81.85 | 79.29 | 78.33 | 80.45 | 90.78 | 85.22 |
| | LLaVA-v1.5-7B-Asia | 84.42 | 81.25 | 88.76 | 81.27 | 82.90 | 83.77 |
| | LLaVA-v1.5-7B-Europe | 84.12 | 80.52 | 80.20 | 90.43 | 83.15 | 81.74 |
| | LLaVA-v1.5-7B-Oceania | 73.78 | 73.16 | 72.38 | 73.91 | 75.03 | 81.16 |
| Scene Understanding | LLaVA-v1.5-7B-Africa | 91.29 | 93.63 | 91.06 | 90.87 | 91.41 | 91.41 |
| | LLaVA-v1.5-7B-America | 92.70 | 93.40 | 92.24 | 91.85 | 94.81 | 91.69 |
| | LLaVA-v1.5-7B-Asia | 94.16 | 94.44 | 95.11 | 92.92 | 94.52 | 92.52 |
| | LLaVA-v1.5-7B-Europe | 93.26 | 92.71 | 92.64 | 94.08 | 93.45 | 92.52 |
| | LLaVA-v1.5-7B-Oceania | 87.54 | 86.57 | 88.08 | 86.55 | 88.11 | 90.58 |

Table 5: Accuracy across different continents for each fine-tuned model.

| Task | Model | Total | Cultural Heritage and Traditions | History and Landmarks | Natural Environment and Local Resources |
|---|---|---|---|---|---|
| Image Recognition | LLaVA-v1.5-7B-CHT | 77.91 | 88.58 | 74.36 | 77.48 |
| | LLaVA-v1.5-7B-HL | 84.76 | 81.57 | 86.89 | 80.25 |
| | LLaVA-v1.5-7B-NELR | 73.46 | 74.89 | 69.18 | 89.88 |
| Cultural Knowledge | LLaVA-v1.5-7B-CHT | 83.29 | 88.53 | 81.71 | 82.16 |
| | LLaVA-v1.5-7B-HL | 87.99 | 84.91 | 90.58 | 82.03 |
| | LLaVA-v1.5-7B-NELR | 81.17 | 80.71 | 79.91 | 86.84 |
| Scene Understanding | LLaVA-v1.5-7B-CHT | 92.96 | 95.27 | 92.45 | 91.73 |
| | LLaVA-v1.5-7B-HL | 94.77 | 95.23 | 94.90 | 93.62 |
| | LLaVA-v1.5-7B-NELR | 91.85 | 92.58 | 91.16 | 93.62 |

Table 6: Accuracy across different categories for each fine-tuned model.

| Dataset | Model | Base | Finetuned |
|---|---|---|---|
| ScienceQA | LLaVA 7B | 70.12 | 69.28 |
| | LLaVA 13B | 74.91 | 75.04 |
| | LLaMA 3.2 11B | 88.97 | 89.30 |
| TextVQA | LLaVA 7B FT | 58.32 | 57.89 |
| | LLaVA 13B FT | 61.18 | 61.13 |
| | LLaMA 3.2 11B | 71.34 | 70.67 |

Table 7: Performance of models before and after fine-tuning on general VQA datasets. *Base* refers to the original model, while *Finetuned* refers to the model adapted using our CultureVerse. The comparable performance across both versions suggests that finetuning on our dataset preserves the models' natural language understanding and commonsense reasoning abilities.

# D DETAILS OF HUMAN ANNOTATION

It is important to note that while CultureVerse's construction pipeline leverages LLMs (e.g., GPT-4o) as an auxiliary tool, we emphasize the optimal synergy between LLM assistance and human verification. For cultural concepts, our primary source is Wikipedia documents, where LLMs only perform basic term extraction. In image collection, while initial retrieval may include irrelevant results, human annotators rigorously filter and curate the final selections. LLMs excel at scalable data retrieval/generation, but human experts ensure correctness through refinement and filtering. Such *balanced human-AI collaboration* is a key factor in CultureVerse's high-quality dataset construction.

## D.1 STATISTICS OF HUMAN ANNOTATORS AND THE PROCESS

In the early stage, our LLM-based filtering step has already removed the majority of low-quality concept–image and QA pairs. Approximately 40% of the initially extracted concepts were filtered out, as they were either culturally irrelevant or difficult to translate into visual cultural concepts; the filtering rates for images and QA pairs were 21% and 18%, respectively. Given that CultureVerse covers nearly 200 countries and regions, it is **nearly impossible to recruit native annotators from all these areas for data annotation**. Therefore, we endeavored to enhance the information retrieval awareness of human annotators to facilitate the labeling process. We conducted a one-week annotation training session for the annotators, utilizing online search engines such as *Google Search* and *Wikipedia*, along with the *Wiki documents* we provided, to verify the test set. This ensured that each data point was annotated and supported by evidence from at least three distinct sources.

Table 8 shows the statistics of human annotators in our study. In total, through the contractor company, we hired 10 expert annotators whose ages are between 20 and 36 with at least a bachelor's degree. Most of them are within the non-AI areas such as education, specific languages, and history. When assigning the annotation job, we asked each annotator to label the correctness, consistency, and relatedness of our questions and answers. Specifically, correctness refers to the correctness of the generated questions and answers, consistency refers to the consistency between the questions, answers, and the concepts, and relatedness aims to make sure that the concepts and generated questions are related to each other. Each instance is labeled by two experts and then verified by another to ensure correctness and consistency. All annotation operations are performed following local laws and regulations to ensure fairness, equity, and accountability.

| Age | % | Degree | % | Major | Language / Culture |
|---|---|---|---|---|---|
| 20-25 | 50% | Bachelor | 50% | Education; Specific languages; Computer science; | English; Chinese |
| 26-36 | 50% | Master | 50% | Communication; Public relation; History | |

Table 8: Statistics of the human annotators to validate CultureVerse.

## D.2 ACCURACY OF HUMAN ANNOTATION

Table 9 shows the precision of human annotators in our generated questions. We then filter out all the wrong questions and only retain the correct ones. It is surprising that automatically generated questions can achieve high accuracy, indicating the promising future of the adoption of advanced AI models like GPT-4o for data collection and annotation.

| Check Item | Accuracy |
|---|---|
| Concept-Region Alignment | 98.25% |
| Concept-Image Alignment | 99.49% |
| Image Recognition Question-Answer Correctness | 98.61% |
| Cultural Knowledge Question-Answer Correctness | 96.52% |
| Scene Understanding Question-Answer Correctness | 93.18% |

Table 9: Accuracy of CultureVerse based on the human annotations.

# E    CASE STUDY

Below, we present examples from CultureVerse representing three different countries.

---

**Concept:** Peking Opera
**Country:** China
**Category:** Music and Dance

**Image Recognition:**
**Question:** What traditional Chinese performance is shown in this image?
**Options:** (A) Henan Opera (B) Peking Opera (C) Kunqu Opera (D) Yue Opera
**Ground truth: (B) Peking Opera**

**Cultural Knowledge:**
**Question:** Which musical instrument is traditionally associated with accompanying the theatrical art form depicted in the image?
**Options:** (A) Guzheng (B) Pipa (C) Jinghu (D) Dizi
**Ground truth: (C) Jinghu**

**Scene Understanding:**
**Question:** In a theatrical performance known for its vibrant costumes and symbolic gestures, what might the intricate face paint and elaborate attire of a character symbolize?
**Options:** (A) The character's age and wisdom (B) The character's social status and role (C) The weather conditions in the story (D) The importance of technology in the narrative
**Ground truth: (B) The character's social status and role**

---

**Concept:** Florence Cathedral
**Country:** Italy
**Category:** Famous Landmarks

**Image Recognition:**
**Question:** What famous cathedral is shown in this image?
**Options:** (A) St. Peter's Basilica (B) Milan Cathedral (C) Florence Cathedral (D) Siena Cathedral
**Ground truth: (C) Florence Cathedral**

**Cultural Knowledge:**
**Question:** Which renowned architect was responsible for engineering the innovative dome of the structure shown in the image?
**Options:** (A) Leon Battista Alberti (B) Filippo Brunelleschi (C) Giorgio Vasari (D) Michelangelo Buonarroti
**Ground truth: (B) Filippo Brunelleschi**

**Scene Understanding:**
**Question:** Imagine you are visiting the city depicted in the image during the Renaissance period. You hear local legends about a stone head on a famous structure there. According to the legend, what was the purpose of this stone head?
**Options:** (A) To ward off evil spirits (B) To grant wishes to passersby (C) To commemorate a donor who contributed a bell (D) To mark the entrance for pilgrims
**Ground truth: (C) To commemorate a donor who contributed a bell**

**Concept:** Pha That Luang
**Country:** Lao People's Democratic Republic
**Category:** Famous Landmarks

**Image Recognition:**
**Question:** What famous structure is shown in this image?
**Options:** (A) Wat Arun (B) Angkor Wat (C) Pha That Luang (D) Shwedagon Pagoda
**Ground truth:** (C) Pha That Luang

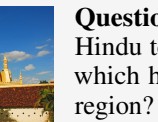

**Cultural Knowledge:**
**Question:** The structure shown in the image, originally built as a Hindu temple, underwent major reconstruction during the reign of which historical figure when Buddhism gained prominence in the region?
**Options:** (A) King Jayavarman VII (B) King Setthathirath (C) King Anawrahta (D) King Ramkhamhaeng
**Ground truth:** (B) King Setthathirath

**Scene Understanding:**
**Question:** Imagine you are attending a festival at the site shown in the image, which is considered one of the most significant Buddhist celebrations in the country. During this event, which of the following cultural practices would you most likely observe that emphasizes the religious and national significance of this site?
**Options:** (A) A parade featuring traditional music and dance (B) A ceremony honoring ancient Hindu deities (C) A cooking contest of traditional Lao dishes (D) A fireworks display celebrating the lunar new year
**Ground truth:** (A) A parade featuring traditional music and dance

## F   PROMPT LIST

```
Concept: {concept}
Country: {country}
Class: {class_}

Please determine whether "{concept}" is a kind of {class_} that can
reflect {country} culture and whether it can serve as a symbol of {
country} culture (unique and very famous within {country}, and not
commonly seen in other parts of the world).
Additionally, the symbol should not be a broad category that includes
various specific items, but rather a distinct and indivisible entity,
such as a specific dance form, a famous individual's photograph, a
renowned landmark, etc.

Here are some counterexamples:
- Broad concepts like "fast food" (which includes burgers, fries, etc.),
"traditional Chinese instruments" (which include guzheng, erhu, etc.) "
Dance" (which include jazz dance, Square dancing, etc.) are not
acceptable due to their lack of a unified visual marker.
- Concept itself is the wrong word, such as "...", "N/A".
- Concept exists in many regions, such as "pork", "duck" or "grapes" (
which might be a specialty or staple in certain countries but is quite
common in many places. However, specific concepts like "peking duck" or "
schweinshaxe" would be correct).
```

```
Please provide your short explanation and include your answer (Yes/No)
into <<<>>>. For example, if you think "{concept}" is a specific and
indivisible symbol of {country} culture, please write <<<Yes>>>.
```

1: Prompt of concept judgement.

```
Please extract cultural elements from the given wikipedia document that
can represent {country} culture or are very famous in {country},
including the following categories:

### Categories
- Food: e.g., local specialty dishes, traditional festival foods.
- Plants: e.g., unique flowers, crops in {country}.
- Animals: e.g., unique wild animals, livestock in {country}.
- Famous Landmarks: e.g., famous historical sites, buildings.
- Festivals: e.g., unique festival celebration scenes.
- Historical Artifacts: e.g., museum collections, ancient relics.
- Historical Figures: e.g., portraits of historical figures, statues.
- Traditional Clothing: e.g., ethnic clothing, festival attire.
- Architectural Styles: e.g., traditional architecture, modern landmark
buildings.
- Handicrafts and Artifacts: e.g., ethnic handicrafts, traditional
handmade items.
- Music and Dance: e.g., traditional musical instruments, dance scenes.
- Religion and Belief: e.g., temples and churches, religious ceremonies.
- Natural Scenery and Ecosystems: e.g., unique natural landscapes,
ecological reserves.
- Markets and Shopping Traditions: e.g., local markets, specialty shops.
- Entertainment and Performing Arts: e.g., theater performances, street
performers.

Please note that not all categories may be included in the document. Only
 list the most famous cultural elements, with a total not exceeding 10.
For categories without famous elements, use 'NA' to indicate. Directly
output in the format "Category: [Element1, Element2, ...]", for example:

### Cultural Elements
- Food: [Food 1], [Food 2], ...
- Plants: NA
- Music and Dance: [Musical Instrument 1], [Dance Scene 2], ...

The following is the wikipedia document.
### Wikipedia Document
{wikipedia}
```

2: Prompt for extracting cultural concept entities from Wikipedia documents.

```
This is the {concept} of {country}. Your task is to generate a multiple-
choice question that asks the user to identify what is shown in the image
. Use the following format for your question:
- Question: [Your Question] Options: (A) [Option 1] (B) [Option 2] (C) [
Option 3] (D) [Option 4]

For example, if the image shows a Peking Duck of China, the question and
options should look like this:
- Question: What traditional dish is shown in this image? Options: (A)
Cantonese Roast Duck (B) Peking Duck (C) Sichuan Spicy Duck (D) Nanjing
Salted Duck

If the image shows the Erhu of China:
- Question: What musical instrument is shown in this image? Options: (A)
Pipa (B) Erhu (C) Sanxian (D) Yangqin

If the image shows the White House of US:
```

```
- Question: What famous American building is shown in this image? Options
: (A) The Capitol (B) The White House (C) The Lincoln Memorial (D) The
Supreme Court Building

Ensure that "{concept}" should be included in one of the options. Ensure
that the options are plausible but only one is the correct answer. The
incorrect options should be similar enough to the correct one to create a
 challenge, but not so similar that they cause any potential ambiguity.
```

3: Prompt for generating scene recognition questions.

```
Please provide a detailed introduction of {concept} of {country},
including information such as: Location and Features (where it is found
or originates from, what makes it unique); Time period (when it was
created or became significant); History (historical background and
development, or any significant events); Cultural significance (cultural
Context in {country}, modern-day significance); Stories or Legends (Any
stories, legends, or folklore associated with {concept}).

Use the following format for your introduction:
Introduction of {concept}: [Detailed Introduction of {concept}]

Here are two examples:

Introduction of Peking Duck: Peking Duck is a famous Chinese dish that
originated in Beijing during the Imperial era. The dish dates back to the
 Yuan Dynasty (1271-1368) and became a staple in the Ming Dynasty
(1368-1644). Traditionally, Peking Duck is known for its thin, crispy
skin and is served with pancakes, hoisin sauce, and scallions. The
preparation involves inflating the duck to separate the skin from the fat
, marinating it, and roasting it in a closed or hung oven. It is
considered a national dish of China and a symbol of Chinese culinary art.

Introduction of The White House: The White House, located at 1600
Pennsylvania Avenue NW in Washington, D.C., is the official residence and
 workplace of the President of the United States. Construction began in
1792 and was completed in 1800. The building was designed by Irish-born
architect James Hoban in the neoclassical style. It has been the
residence of every U.S. president since John Adams. The White House has
undergone several renovations and expansions, including the addition of
the West Wing and the Oval Office. It is a symbol of the U.S. government
and a site of significant historical events.

Now please provide the introduction for {concept}.
Introduction of {concept}:
```

4: Prompt for generating the introduction of cultural concept.

```
This public image shows the "{concept}" of {country}. Generate a multiple
-choice question based on this image and the introduction of {concept}.
Provide the correct answer immediately following the question.

Ensure the question delves into deeper cultural knowledge but does not
directly name the {concept}. The options should be somewhat confusing to
increase the difficulty, but there must be only one correct answer. Users
 can only answer based on the image, so don't mention any "introduction"
or "{concept}" in the question. Use the following format for your
generated question:
- Question: [Your Question] Options: (A) [Option 1] (B) [Option 2] (C) [
Option 3] (D) [Option 4]
- Answer: (X) [Option X]

Here are two examples:
```

```
Image: Peking Duck
Introduction of Peking Duck: Peking Duck is a famous Chinese dish that
originated in Beijing during the Imperial era. The dish dates back to the
 Yuan Dynasty (1271-1368) and became a staple in the Ming Dynasty
(1368-1644). Traditionally, Peking Duck is known for its thin, crispy
skin and is served with pancakes, hoisin sauce, and scallions. The
preparation involves inflating the duck to separate the skin from the fat
, marinating it, and roasting it in a closed or hung oven. It is
considered a national dish of China and a symbol of Chinese culinary art.

- Question: During which dynasty did the dish shown in the image become a
 staple in the cuisine of its country? Options: (A) Tang Dynasty (B) Song
 Dynasty (C) Ming Dynasty (D) Qing Dynasty
- Answer: (C) Ming Dynasty

Image: The White House
Introduction of The White House: The White House, located at 1600
Pennsylvania Avenue NW in Washington, D.C., is the official residence and
 workplace of the President of the United States. Construction began in
1792 and was completed in 1800. The building was designed by Irish-born
architect James Hoban in the neoclassical style. It has been the
residence of every U.S. president since John Adams. The White House has
undergone several renovations and expansions, including the addition of
the West Wing and the Oval Office. It is a symbol of the U.S. government
and a site of significant historical events.

- Question: Who was the architect responsible for designing the building
shown in the image? Options: (A) James Hoban (B) Benjamin Latrobe (C)
Thomas Jefferson (D) Charles Bulfinch
- Answer: (A) James Hoban

Now please generate the question for the Image: {concept} of {country}
{introduction}
```

5: Prompt for generating cultural knowledge questions based on the introduction of cultural concepts.

```
This public image shows the "{concept}" of {country}. Generate a visual
reasoning multiple-choice question based on this image and the
introduction of {concept}. Provide the correct answer and reason
immediately following the question.

Here are some requirements:
- The question must describe a specific scenario crafted to test deeper
cultural understanding without directly naming {concept}. The scenario
can be related to cultural background, regional characteristics,
historical legends, or etiquette and customs, etc.
- The question needs to be related to the image but does not need to
describe the content of the image.
- Ensure the question requires the user to recognize the image and use
relevant knowledge to answer through reasoning based on the scenario
provided. Users can only answer based on the image, so don't mention any
"introduction" or "{concept}" in the question.

Use the following format for your introduction and question:
- Question: [Your Scenario-based Question] Options: (A) [Option 1] (B) [
Option 2] (C) [Option 3] (D) [Option 4]
- Answer: (X) [Option X]
- Reason: [Your Reason for the Answer]

Here are two examples:

Image: Peking Duck
```

```
Introduction of Peking Duck: Peking Duck is a famous Chinese dish that
originated in Beijing during the Imperial era. The dish dates back to the
 Yuan Dynasty (1271-1368) and became a staple in the Ming Dynasty
(1368-1644). Traditionally, Peking Duck is known for its thin, crispy
skin and is served with pancakes, hoisin sauce, and scallions. The
preparation involves inflating the duck to separate the skin from the fat
, marinating it, and roasting it in a closed or hung oven. This
meticulous process ensures the skin becomes crispy while the meat remains
 tender. Peking Duck is often carved in front of diners and served in
three stages: the skin, the meat, and a broth made from the bones. It is
considered a national dish of {country} and a symbol of Chinese culinary
art. Peking Duck has also been a part of many state banquets and
diplomatic events, symbolizing Chinese hospitality and culinary
excellence.

- Question: During a state banquet featuring the dish in the image, which
  aspect of its presentation is most likely emphasized to symbolize
Chinese culinary excellence and hospitality? Options: (A) The use of
exotic spices (B) The serving of the duck with rice (C) The incorporation
 of seafood (D) The carving of the duck in front of diners
- Answer: (D) The carving of the duck in front of diners
- Reason: The traditional carving of Peking Duck in front of diners
highlights the skill involved in its preparation and serves as a symbol
of Chinese culinary excellence and hospitality.

Image: The White House
Introduction of The White House: The White House, located at 1600
Pennsylvania Avenue NW in Washington, D.C., is the official residence and
 workplace of the President of the United States. Construction began in
1792 and was completed in 1800. The building was designed by Irish-born
architect James Hoban in the neoclassical style, featuring a white-
painted Aquia Creek sandstone exterior. It has been the residence of
every U.S. president since John Adams. The White House has undergone
several renovations and expansions, including the addition of the West
Wing, East Wing, and the Oval Office. The building's iconic appearance
and historical significance make it a symbol of the U.S. government and a
 site of significant historical events. The White House has been the
location of many important decisions, meetings with foreign dignitaries,
and addresses to the nation. It also serves as a museum of American
history, housing numerous artifacts and pieces of art. The White House is
 not only a residence but also a working office, with various staff
members ensuring the smooth operation of the executive branch of the U.S.
  government.

- Question: During a critical diplomatic event, the President is
scheduled to meet with several foreign dignitaries to discuss global
climate initiatives. As depicted in the image, which room inside the
building is most likely to be used for this high-level diplomatic meeting
? Options: (A) The Lincoln Bedroom (B) The Oval Office (C) The White
House Kitchen (D) The East Room
- Answer: (B) The Oval Office
- Reason: The Oval Office is traditionally used for important meetings
and discussions, making it the most likely choice for a high-level
diplomatic meeting with foreign dignitaries.

Now please generate the question for the Image: {concept} of {country}
{introduction}
```

6: Prompt for generating scene reasoning questions based on the introduction of cultural concepts.

```
(Hint: This image shows the {concept} of {country}.)
```

```
Here is a question about this image:
{question}

First, describe the image in detail and analyze its features. Then,
analyze the characteristics of the four options and compare each one with
 the features of the image. Finally, provide your final answer. Please
include your answer into (). For example, if you choose A, please write (
A).
```

7: Prompt for generating knowledge-based reasoning response to image recognition questions in training set.

```
(Hint: This image shows the {concept} of {country}. {introduction})

Here is a question about this image:
{question}

First, provide a detailed description and identification of the image,
analyzing its features. Then, conduct a comprehensive analysis of the
question and four options based on the Hint and your knowledge. Finally,
present the final answer.

Please refrain from explicitly mentioning the "Hint" in your response, as
 these are for your discreet knowledge and not provided by the question.
Please include your answer into (). For example, if you choose A, please
write (A).

Response:
```

8: Prompt for generating knowledge-based reasoning response to cultural knowledge / scene reasoning questions in training set.

```
Here is a question about this image:
Question: {question} Options: {options}

First, describe and identify the image. Then, analyze the question and
all four options in detail. Finally, provide the answer, indicating your
final choice in parentheses. For example, if you choose A, please write (
A).
```

9: Prompt for stepwise reasoning in the evaluation of CultureVerse.

```
# Simplified Annotation Guidelines
## Overview

Each data entry consists of a Cultural Concept, a Country, an Image, and
1 to 3 Multiple-Choice Questions. Your task is to evaluate the quality of
 each entry by completing 5 judgments (Judge 1 - Judge 5).

For data that can be corrected, perform annotations and modifications to
ensure complete accuracy.

For data that is irreversible, unmodifiable, or unverifiable, mark it for
 deletion.

## Mandatory Tools

Search Engine Verification: Use Google, Wikipedia, or other reliable
sources to verify all data. Do not rely solely on personal knowledge.

Cross-Referencing: Confirm uncertain cultural facts or historical dates
with at least three different sources.
```

```
## Annotation Steps
### Phase 1: Concept & Image Verification (Judge 1 & Judge 2)

1. [Judge 1] Concept Validity

Criteria: The Concept must be correctly spelled and be culturally
representative of (or unique to) the specified Country.

2. [Judge 2] Image Relevance

Criteria: The Image must clearly depict the cultural concept, possess
high visual quality, and be free of sensitive, offensive, or irrelevant
content.

### Phase 2: QA Pair Verification (Judge 3 – Judge 5)

Proceed only if Judge 1 and Judge 2 have passed. Review each Question (Q1
, Q2, Q3) individually.

3. [Judge 3 / 4 / 5] Question & Answer Quality

Criteria:

- Question: The question must be grammatically correct, logical, and
relevant to the provided image and concept.
- Answer: The correct answer must be factually accurate according to
external search results.
- Options: All four options must be distinct and mutually exclusive.
Ensure there are no duplicate correct answers or ambiguous distractors.
```

10: Manual Annotation Guidelines (Simplified Version)

| Country | Concept | Country | Concept | Country | Concept |
|---|---|---|---|---|---|
| **SUM of Concepts** | **19,682** | Sri Lanka | 90 | Samoa | 30 |
| India | 1,430 | Democratic People's Republic of Korea | 90 | Turkmenistan | 27 |
| United States of America | 1,411 | Saudi Arabia | 87 | Qatar | 26 |
| Italy | 661 | Lithuania | 86 | Guyana | 26 |
| China | 545 | Malta | 86 | Kuwait | 25 |
| Mexico | 524 | Uzbekistan | 84 | Paraguay | 24 |
| Japan | 522 | Algeria | 83 | Sudan | 24 |
| Philippines | 465 | Lebanon | 79 | Angola | 24 |
| Indonesia | 400 | Nigeria | 78 | Fiji | 24 |
| France | 374 | Colombia | 78 | Seychelles | 23 |
| Russian Federation | 328 | Austria | 77 | Lesotho | 22 |
| Greece | 300 | Cyprus | 75 | Barbados | 21 |
| Germany | 273 | Mongolia | 74 | Dominica | 21 |
| Egypt | 272 | Cuba | 73 | Mauritius | 20 |
| Armenia | 259 | Bosnia and Herzegovina | 72 | Maldives | 19 |
| Australia | 254 | Ecuador | 71 | Niger | 18 |
| Spain | 249 | Slovakia | 65 | Zambia | 18 |
| Georgia | 245 | Iceland | 65 | Antigua and Barbuda | 17 |
| Brazil | 239 | Luxembourg | 65 | Saint Lucia | 16 |
| Canada | 231 | Iraq | 62 | Eswatini | 16 |
| Thailand | 228 | Ghana | 61 | Bahrain | 15 |
| Myanmar | 226 | Albania | 60 | Papua New Guinea | 15 |
| Ireland | 220 | Uruguay | 60 | Kazakhstan | 15 |
| Pakistan | 218 | Montenegro | 60 | Tuvalu | 14 |
| Nepal | 216 | Uganda | 58 | Liechtenstein | 14 |
| New Zealand | 209 | Chile | 56 | Côte d'Ivoire | 14 |
| Portugal | 199 | Senegal | 56 | Suriname | 13 |
| Ukraine | 197 | United Republic of Tanzania | 56 | Bahamas | 12 |
| Malaysia | 189 | United Arab Emirates | 56 | Eritrea | 12 |
| Bangladesh | 188 | Guatemala | 54 | Mozambique | 12 |
| Peru | 188 | Latvia | 53 | Burundi | 11 |
| Poland | 186 | Yemen | 52 | Marshall Islands | 11 |
| Bulgaria | 182 | Afghanistan | 51 | Honduras | 11 |
| United Kingdom of Great Britain and Northern Ireland | 176 | Belarus | 50 | San Marino | 11 |
| Croatia | 176 | Benin | 50 | Liberia | 11 |
| Serbia | 172 | Oman | 50 | Tajikistan | 11 |
| Romania | 167 | Dominican Republic | 49 | Solomon Islands | 10 |
| Ethiopia | 155 | Tunisia | 48 | Comoros | 9 |
| Cambodia | 154 | Trinidad and Tobago | 48 | Nauru | 9 |
| Netherlands | 151 | El Salvador | 47 | Vanuatu | 8 |
| Denmark | 142 | Monaco | 47 | Kiribati | 8 |
| Lao People's Democratic Republic | 141 | Mali | 46 | Timor-Leste | 8 |
| South Africa | 138 | Kenya | 45 | Grenada | 8 |
| Argentina | 134 | Estonia | 45 | Sierra Leone | 7 |
| Republic of Korea | 132 | Haiti | 44 | Rwanda | 7 |
| Czechia | 130 | Jamaica | 43 | Guinea | 7 |
| Bhutan | 130 | Belize | 42 | Libya | 7 |
| Azerbaijan | 129 | Plurinational State of Bolivia | 41 | Andorra | 7 |
| Morocco | 126 | Congo | 40 | Gambia | 7 |
| Norway | 120 | Namibia | 39 | Burkina Faso | 7 |
| Sweden | 112 | Somalia | 39 | South Sudan | 5 |
| Finland | 112 | Nicaragua | 38 | Gabon | 5 |
| Israel | 109 | Kyrgyzstan | 36 | Togo | 4 |
| Türkiye | 109 | Bolivarian Republic of Venezuela | 35 | Malawi | 4 |
| Viet Nam | 109 | Madagascar | 35 | Saint Kitts and Nevis | 4 |
| Hungary | 101 | Republic of Moldova | 34 | Djibouti | 4 |
| Ethnic_and_religiou_groups | 94 | Zimbabwe | 33 | Saint Vincent and the Grenadines | 3 |
| North Macedonia | 94 | Jordan | 33 | Central African Republic | 3 |
| Slovenia | 93 | Tonga | 32 | Chad | 2 |
| Islamic Republic of Iran | 93 | Botswana | 32 | Mauritania | 2 |
| Switzerland | 93 | Cameroon | 31 | Panama | 1 |
| Singapore | 93 | Costa Rica | 31 | Federated States of Micronesia | 1 |
| Belgium | 91 | Democratic Republic of the Congo | 31 | Equatorial Guinea | 1 |

Table 10: Number of cultural concepts of different countries or regions.

