# OpenReview forum: "CultureVLM: Characterizing and Improving Cultural Understanding of Vision-Language Models"
_ICLR.cc/2026/Conference — ICLR 2026 Conference Withdrawn Submission_

### Official Review · Reviewer_2WM5 · 2025-10-31

**Soundness:** 3
**Presentation:** 4
**Contribution:** 3
**Rating:** 6
**Confidence:** 4

**Summary:**

This paper presents CultureVerse a scalable pipeline and multimodal multiple choice corpus covering 188 countries/regions, 15 cultural topics and 3 question types. The proposed corpus is obtained using GPT-4o to process Wikipedia Documents and extract cultural concepts, afterwards images are retrieved from Google Images, and finally QA pairs are generated, containing 3 QA types: Image, Recognition, Cultural Knowledge and Scene Understanding. The authors also propose CultureVLM, which are VLMs that are finetuned on the proposed resource (CultureVerse) with the overall goal of improving the global cultural perception and understanding. The paper shows evaluations with multiple VLMs, showing that finetuning on CultureVerse improves the cultural understanding of the models, the authors perform multiple evaluations showing regional disparities, the relation of cultural understanding and model/data scale, generalization of cultural concepts and more.

**Strengths:**

Paper Strenghs:
1. The authors tackle the gap of the limited understanding of culture in VLMs, with a scalable method to gather data from different cultures and regions, which has been an issue in past works. Another strength is the creation of different types of questions to evaluate and improve the multicultural knowledge of VLMs from different aspects.
2. Finetuning Models in CultureVerse helps the models to improve their cultural perception and cross-cultural generalization. The authors show improvements in different datasets such as CVQA and CulturalVQA showing improvements on both.
3. The paper was well written and easy to read with broad evaluations and ablations.

**Weaknesses:**

Paper Weaknesses:
1. In section 3.1 - Image Retrieval, it is not really clear how the images were obtained, this was done manually or using directly some tool with Google Images? . On the other hand, the authors state that for each cultural concept 5 images are obtained, in my personal experience finding images specially for low-resource cultures is hard, so maybe the authors can clarify a bit more on this process. Also, how did you managed the situation with licenses, many images on the web are not free. (Which increase the difficulty of finding images related to a certain aspect of a culture), did you use specific webpages ?
2. I think that a weakness is the human evaluation, the paper states that "Annotators refined the questions and answers by removing redundant information and resolving any ambiguities to maintain clarity and accuracy". I really think that you need specialized annotators for this, judging aspects of other cultures is hard for fixing QA pairs (even if you use google search), that is why works like CVQA works with people from all around the world for this type of validations, however I understand this limits scalability. On the other hand, the paper states that only 10 annotators were hired to do all this process, and due to the scale of the proposed corpus it's not really clear how this task was managed by only 10 annotators, which brings concerns of the quality of the validation. I would like to see the authors' point of view on this matters.
3. Due to CultureVerse is a multimodal corpus, how does the model works when you do not give the image ?. A well know fact, is that VLMs over-rely on text. In page 20, the authors show examples of the corpus, the questions on Image Recognition and Cultural Knowledge   seems to always refer to the image, which is good, but in the case of Scene Understanding (first example: "In a theatrical performance..."), the question is not directly referring to the image, so the VLM can rely entirely on the text, perform the reasoning and answer the question. If there is enough time, it would be good to see performances when the images are not given to the models, even if is for one model.
4. Only as a suggestion if the authors agree, when I saw figure 2a the first time, it gives the impression that Asia contains only Landmarks, or that cultural knowledge is coming only from Europe. So a better design for this graph could be considered to give a better understanding.

**Questions:**

Please refer to Weaknesses for my question and doubts. Overall I think the paper brings good contributions, so I'm currently giving it borderline accept, but I would like to see the responses of the authors on my concerns, and sorry if I misunderstood something. After Rebuttal I will revise my decision.

---

> ### Author Response · Authors · 2025-11-23
> **Response to Reviewer 2WM5 (1)**
>
> Thank you for your encouraging review and for recognizing the scalability of our data pipeline, the design of diverse QA types for multifaceted cultural evaluation, and the effectiveness of fine-tuning on cross-cultural generalization. We sincerely appreciate your constructive feedback and will address all concerns below to further refine and strengthen our work.
>
> > **Q1:** In section 3.1 - Image Retrieval, it is not really clear how the images were obtained, this was done manually or using directly some tool with Google Images? . On the other hand, the authors state that for each cultural concept 5 images are obtained, in my personal experience finding images specially for low-resource cultures is hard, so maybe the authors can clarify a bit more on this process. Also, how did you managed the situation with licenses, many images on the web are not free. (Which increase the difficulty of finding images related to a certain aspect of a culture), did you use specific webpages ?
>
> We apologize for the lack of clarity. Regarding image acquisition, we utilized the Google Images retrieval tool for batch collecting. We fully agree that finding high-quality images for low-resource cultures is challenging. Consequently, while our initial target was five images per concept, rigorous filtering and selection meant that **not every concept retained all five images** in the final dataset.
>
> Regarding licensing and privacy, we applied filters based on usage rights during retrieval, blurred sensitive private information, and strictly emphasize that all images are for research purposes only. We have added these specific details to Section 3.1 of the paper to clarify the process.
>
> > **Q2:** I think that a weakness is the human evaluation, the paper states that "Annotators refined the questions and answers by removing redundant information and resolving any ambiguities to maintain clarity and accuracy". I really think that you need specialized annotators for this, judging aspects of other cultures is hard for fixing QA pairs (even if you use google search), that is why works like CVQA works with people from all around the world for this type of validations, however I understand this limits scalability. On the other hand, the paper states that only 10 annotators were hired to do all this process, and due to the scale of the proposed corpus it's not really clear how this task was managed by only 10 annotators, which brings concerns of the quality of the validation. I would like to see the authors' point of view on this matters.
>
> Thank you for this insightful comment. Cultural data annotation is indeed a nuanced and complex challenge, something we deeply appreciated after performing annotations ourselves and managing the hired team. Our experience revealed an important reality: **even when annotating culture from our own countries/regions, the diversity across cities and sub-regions often meant we lacked deep familiarity with specific local concepts**. In many cases, we found that even "native" annotators ultimately had to rely on search engines to verify details. Therefore, being a native speaker does not automatically guarantee a deep understanding of every specific cultural niche within a region.
>
> Consequently, we opted for a rigorous verification process using a dedicated team. We employed 10 annotators (a small, manageable team) who worked for over a month specifically on the 10k concepts in the test subset. They were instructed to cross-reference multiple information sources via search engines for every data point. We believe that managing a smaller, focused team over a longer period allowed for stricter quality control and consistency compared to managing a massive, distributed crowd. We have included these specific details in Appendix D.

---

> > ### Author Response · Authors · 2025-11-23
> > **Response to Reviewer 2WM5 (2)**
> >
> > > **Q3:** Due to CultureVerse is a multimodal corpus, how does the model works when you do not give the image ?. A well know fact, is that VLMs over-rely on text. In page 20, the authors show examples of the corpus, the questions on Image Recognition and Cultural Knowledge seems to always refer to the image, which is good, but in the case of Scene Understanding (first example: "In a theatrical performance..."), the question is not directly referring to the image, so the VLM can rely entirely on the text, perform the reasoning and answer the question. If there is enough time, it would be good to see performances when the images are not given to the models, even if is for one model.
> >
> > Thank you for this constructive suggestion! We acknowledge that VLMs often exhibit a tendency to over-rely on textual cues. To investigate this, we conducted an experiment where the model was tasked with inferring the correct answer based solely on the question and options, without the image input. The results are presented in the table below. As observed, there is a significant performance drop across all tasks. Task 1 (Image Recognition) drops to near-random guessing levels, which is expected as the task is functionally paralyzed without visual input. Task 3 (Scene Understanding) shows a relatively smaller decline; this is likely because the task definition focuses on social appropriateness—avoiding anti-religious or culturally offensive responses—making it somewhat easier to guess the culturally "safe" or appropriate option from text alone. Meanwhile, we also observe that for smaller-scale models, inferring the correct answer from text alone poses a greater challenge. We will collect more detailed statistics and include the corresponding visualizations and discussion in the revised manuscript. Thank you for helping us strengthen our analysis!
> >
> > |  | Task1 | Task2 | Task3 |
> > | --- | --- | --- | --- |
> > | **LLaVA-1.5-7B (Text Only)** | 0.294 | 0.363 | 0.610 |
> > | **LLaVA-1.5-7B (Text + Image)** | 0.512 | 0.481 | 0.724 |
> > | Δ | **-42.6%** | **-24.5%** | **-15.7%** |
> > | **LLaVA-OneVision-0.5B (Text Only)** | 0.241 | 0.216 | 0.359 |
> > | **LLaVA-OneVision-0.5B (Text + Image)** | 0.454 | 0.364 | 0.653 |
> > | Δ |  **-46.9%** |  **-40.7%** |  **-45.0%** |
> >
> > > **Q4:** Only as a suggestion if the authors agree, when I saw figure 2a the first time, it gives the impression that Asia contains only Landmarks, or that cultural knowledge is coming only from Europe. So a better design for this graph could be considered to give a better understanding.
> >
> > We appreciate this detailed suggestion. To prevent potential confusion, we have modified Figure 2a by rotating the pie chart to improve readability. This update has been incorporated into the revised paper. Please check it in the revised manuscript.
> >
> > > **Q5:** Overall I think the paper brings good contributions, so I'm currently giving it borderline accept, but I would like to see the responses of the authors on my concerns, and sorry if I misunderstood something. After Rebuttal I will revise my decision.
> >
> > Thank you for recognizing the recognition the contributions and the effort behind this work! We are also appreciate for your insightful suggestions and the opportunity for discussion. We hope our responses could address your concerns and encourage a positive evaluation of our work, and we also warmly welcome any further questions or discussion.

---

### Official Review · Reviewer_pyUY · 2025-11-01

**Soundness:** 2
**Presentation:** 3
**Contribution:** 3
**Rating:** 6
**Confidence:** 4

**Summary:**

The paper introduces CultureVLM, a series of vision–language models fine-tuned on CultureVerse, a new large-scale multimodal dataset covering 188 countries, 15 cultural topics, and over 19,000 cultural concepts. The dataset includes three types of tasks—image recognition, cultural knowledge, and scene understanding—and aims to measure and improve VLMs’ cross-cultural generalization. Experiments on 16 models show that CultureVLM enhances cultural awareness while maintaining general capabilities on other VQA benchmarks. Analyses reveal regional disparities (Western bias) and demonstrate that fine-tuning on CultureVerse mitigates them, particularly for Asia and Africa.

**Strengths:**

- CultureVerse substantially expands geographic and conceptual coverage beyond existing datasets such as MaRVL, CVQA, CulturalVQA, and ALM-Bench, reaching near-global representation.
- The authors evaluate 16 models across tasks, regions, and cultural categories, offering an unusually systematic view of cross-cultural disparities.
- The paper provides ablations on fine-tuning size, generalization across continents and categories, and catastrophic forgetting, which strengthens the empirical contribution.
- The paper addresses fairness and inclusion in multimodal AI, highlighting performance gaps for low-resource regions.

**Weaknesses:**

- Reliance on Wikipedia and GPT-generated questions. These sources risk reinforcing the same Western-centric biases the paper seeks to mitigate; the authors should better quantify how much content originates from non-Western data and how QA quality varies by region. Moreover, since native annotators from nearly 200 regions were not available, annotators relied on Google and Wikipedia to infer correctness. This may reproduce Western-centric biases and limit true cultural authenticity. The authors could at least acknowledge this limitation or complement it with small-scale native validation for underrepresented regions.

- Evaluation metrics. Accuracy alone may not capture the subtlety of cultural understanding and reasoning. Alternative or human-evaluated metrics could strengthen validity. It would also be great if you could elaborate further on how you evaluate the cultural stepwise reasoning process in the recognition tasks. It seems you only evaluate the model's multiple-choice selection and not the reasoning process. How do you ensure the reasoning process is indicative of the output? Also, in the cultural knowledge questions, you mention that reasoning is required, but I do not see how reasoning is involved in this task. Could you further elaborate and provide an example? There is also an inconsistency, as in lines 374-375, you mention stepwise reasoning as a prompt method for the image recognition task, and in the appendix (lines 859-860), you mention it for cultural knowledge and scene understanding questions.

- Reproducibility and filtering pipeline. The dataset construction process involves multiple stages (entity extraction, LLM question generation, filtering), but the filtering criteria are not well quantified. What proportion of automatically generated samples were discarded or corrected? How were duplicates or culturally ambiguous examples handled? The absence of these statistics limits reproducibility and makes it difficult to assess dataset quality or coverage.

- Dataset transparency and annotation quality. While Appendix D.1 offers a general overview of the annotation process, it omits key quantitative and reproducibility details. The paper should report the cultural background of the annotators, inter-annotator agreement, annotation guidelines, and procedures for logging evidence, and discuss potential Western bias introduced by relying solely on non-native annotators verifying cultural content through search engines and Wikipedia. Recent work has shown that in known computer vision benchmarks where crowdsourced annotators were employed, there were cultural implicatures that were difficult for them to identify and were later extracted by annotators with culture-specific backgrounds [Karamolegkou et al., 2024](https://aclanthology.org/2024.hucllm-1.5/).

**Questions:**

- How do you ensure that the QA pairs generated by GPT-4o reflect authentic cultural knowledge rather than textual stereotypes? Was any cross-validation with native annotators performed?

- Can you provide statistics on the proportion of non-English Wikipedia documents or local-language sources included per region?

- Since “scene understanding” involves context-sensitive reasoning, how is ambiguity resolved when multiple answers might be culturally plausible?

**Comments**

- In Figure 4, you should specify in the caption what performance (%) indicates. You should also specify what the readers see in a, b, c, d.

- In lines 859-861, you mention that you use stepwise reasoning for cultural knowledge and scene understanding questions, while in lines 373-374, you mention that you perform stepwise reasoning in the recognition tasks. Please clarify the prompting setups for each task.

- The authors state that the dataset “will be governed through protocols and licenses” but do not specify which protocols or licenses (e.g., CC-BY, CC-BY-NC-SA, or data-use agreements). Providing this information would clarify how the community can legally and ethically use CultureVerse and ensure compliance with open-science standards.

---

> ### Author Response · Authors · 2025-11-23
> **Response to Reviewer pyUY (1)**
>
> Thank you for your thoughtful review and for recognizing the broad geographic and conceptual coverage of CultureVerse and our empirical analyses on fine-tuning, generalization, and fairness. Here we clarify the questions and address your concerns point by point:
>
> > **Q1:** Reliance on Wikipedia and GPT-generated questions. These sources risk reinforcing the same Western-centric biases the paper seeks to mitigate; the authors should better quantify how much content originates from non-Western data and how QA quality varies by region. Moreover, since native annotators from nearly 200 regions were not available, annotators relied on Google and Wikipedia to infer correctness. This may reproduce Western-centric biases and limit true cultural authenticity. The authors could at least acknowledge this limitation or complement it with small-scale native validation for underrepresented regions.
> > **Q6:** Can you provide statistics on the proportion of non-English Wikipedia documents or local-language sources included per region?
>
> We acknowledge that some degree of residual bias is an inherent challenge in large-scale datasets. However, we have taken extensive measures to mitigate this throughout our pipeline—from source selection and construction to filtering and final annotation. The table below shows the geographic distribution of cultural concepts in our raw data sources, drawn from Wikipedia documents across ~200 countries. It demonstrates comprehensive global coverage, with Asia, Africa, and Oceania together accounting for over 40% of the data, indicating substantial non-Western representation. Furthermore, after our curation process, the final dataset achieves a representation of **49.6%** non-Western content (detailed distributions are visualized in Figure 2).
>
> We would like to emphasize that our dataset focuses primarily on **objective cultural knowledge** (e.g., factual details about clothing, food, and landmarks derived from Wikipedia documents) rather than subjective or opinion-based content. This focus significantly reduces the risk of propagating subjective Western-centric biases. We believe that **our rigorous pipeline, combined with the high proportion of non-Western data, effectively mitigates the majority of potential biases**.
>
> | Continent | Country | Concepts | Percentage |
> | --- | --- | --- | --- |
> | America | 35 | 74,602 | 30.3% |
> | Europe | 43 | 69,459 | 28.2% |
> | Asia | 47 | 73,735 | 30.0% |
> | Africa | 54 | 18,870 | 7.7% |
> | Oceania | 14 | 9,371 | 3.8% |

---

> > ### Author Response · Authors · 2025-11-23
> > **Response to Reviewer pyUY (2)**
> >
> > > **Q2:** Evaluation metrics.
> > > **Q8-2:** In lines 859-861, you mention that you use stepwise reasoning for cultural knowledge and scene understanding questions, while in lines 373-374, you mention that you perform stepwise reasoning in the recognition tasks. Please clarify the prompting setups for each task.
> >
> > We apologize for any potential misunderstanding. Regarding **Cultural Knowledge** and **Scene Understanding** tasks, our default approach is to use a prompt that explicitly requires stepwise reasoning. Specifically, we ask the model to first identify the cultural concept depicted in the image and then analysis about the knowledge behind that concept before selecting an option. We believe this is a more logical approach to answering complex queries. We observed that without this instruction, some models tend to "guess" the final option directly without grounding their answer in the image content. An example of the prompt and LLaVa output  is shown below:
> > ```
> > PROMPT: Here is a question about this image:\\nQuestion: As depicted in the image, people are bathing in the waters at a religious site. What practice is associated with this activity? Options: (A) Meditation in a Buddhist Monastery (B) Washing in the Amazon River (C) Baptism in the Jordan River (D) Bathing in the waters as a part of Hindu rituals\\n\\nFirst, describe and identify the image. Then, analyze the question and all four options in detail. Finally, provide the answer, indicating your final choice in parentheses. For example, if you choose A, please write (A).
> >
> > OUTPUT: The image shows a large group of people gathered in a body of water, possibly a river or a lake, with a temple in the background. The people are washing themselves, and there are umbrellas set up around the area. ... Analysis:\\n(A) Meditation in a Buddhist Monastery: This option is not related to the image, as it involves meditation in a monastery rather than washing in a body of water.\\n(B) Washing in the Amazon River: This option is not related to the image, as it refers to the practice of washing in the Amazon River, which is not the same as the washing in the body of water shown in the image.\\n(C) Baptism in the Jordan River: This option is not related to the image, as it involves the practice of baptism in the Jordan River, which is not the same as the washing in the body of water shown in the image.\\n(D) Bathing in the waters as a part of Hindu rituals: This option is related to the image, as the people are washing themselves in the body of water, which is a common practice in Hinduism. Hindu rituals often involve bathing in sacred rivers or lakes to cleanse sins and lead to liberation.\\n\\nAnswer: (D) Bathing in the waters as a part of Hindu rituals
> > ```
> > Conversely, for **Image Recognition** tasks, our default setting uses a direct prompt without explicit stepwise reasoning steps. This is because image recognition is primarily a perceptual task—directly identifying a concept—rather than a multi-step reasoning task. As analyzed in Section 4.2 (and illustrated in Figure 8), our experiments showed that enforcing stepwise reasoning for simple recognition tasks did not yield performance improvements and, in some cases, even had negative impact.
> >
> >
> > > **Q3:** Reproducibility and filtering pipeline. The dataset construction involves multiple stages, but key filtering details—such as the proportion of discarded or corrected samples and how duplicates or culturally ambiguous examples were handled—are missing, limiting reproducibility and quality assessment.
> > > **Q7:** Since “scene understanding” involves context-sensitive reasoning, how is ambiguity resolved when multiple answers might be culturally plausible?
> >
> > Thank you for the constructive feedback. Actually, we have documented the primary criteria for the multi-stage pipeline (entity extraction, generation, and filtering) in the form of prompts within Appendix F. More granular technical details will be released with our code. Approximately 40% of the initially extracted concepts were filtered out, as they were either culturally irrelevant or difficult to translate into visual cultural concepts; the filtering rates for images and QA pairs were 21% and 18%, respectively. More details will be released along with the code.
> >
> > To better illustrate our filtering standards and how we handle ambiguity, we have also included a simplified version of our **Human Annotation Guidelines** in the revised manuscript. These can now be found in **Appendix F, Prompt 10**.
> >
> > We address potential ambiguity through a two-stage verification process. During the LLM filtering stage, we automatically eliminate QA pairs that contain confusing or multiple plausibly correct answers. During the final human annotation stage, we explicitly instruct annotators to ensure the uniqueness of the ground truth. They are required to either refine the answer to be distinct or discard the QA pair entirely if the ambiguity cannot be resolved.

---

> > > ### Author Response · Authors · 2025-11-23
> > > **Response to Reviewer pyUY (3)**
> > >
> > > > **Q4:** Dataset transparency and annotation quality. While Appendix D.1 offers a general overview of the annotation process, it omits key quantitative and reproducibility details. The paper should report the cultural background of the annotators, inter-annotator agreement, annotation guidelines, and procedures for logging evidence, and discuss potential Western bias introduced by relying solely on non-native annotators verifying cultural content through search engines and Wikipedia. Recent work has shown that in known computer vision benchmarks where crowdsourced annotators were employed, there were cultural implicatures that were difficult for them to identify and were later extracted by annotators with culture-specific backgrounds Karamolegkou et al., 2024.
> > >
> > > Thank you for the insightful comments on our annotation process. We have revised Appendix D/F to include detailed information such as annotator backgrounds, language, and annotation guidelines. More details will be shown in the code.
> > >
> > > We would also like to discuss our rationale regarding the test subset annotation. We acknowledge that crowdsourced annotators may inadvertently introduce their own cultural biases, as noted in your referenced paper. However, finding perfectly matched culture-specific annotators for nearly 200 regions is extremely challenging. In our own pilot studies, we found that even when annotating culture from our own countries, regional variances often meant we lacked deep familiarity with specific local concepts from other cities or sub-regions. Thus, even "native" annotators who are not from the exact locale can introduce bias. Consequently, we prioritized evidence-based verification. By mandating that annotators cross-reference multiple information sources via search engines, we aim to minimize subjective bias and rely on verifiable facts. We believe that managing a small, focused team over a longer period allowed for stricter quality control and consistency compared to managing a massive, distributed crowd of varied backgrounds. **Cultural data annotation is indeed a nuanced and complex challenge**—a reality we deeply appreciated after performing annotations ourselves and managing the team.
> > >
> > > > **Q5:** How do you ensure that the QA pairs generated by GPT-4o reflect authentic cultural knowledge rather than textual stereotypes? Was any cross-validation with native annotators performed?
> > >
> > > To minimize the risk of hallucinations or stereotypes, our QA pairs are primarily extracted directly from Wikipedia documents by GPT-4o, rather than being generated from the model's internal priors. This ensures the content reflects documented, objective cultural knowledge. Furthermore, as authors, we conducted a verification of the data related to our own native cultures, finding that the error rate remained below 1%. This internal cross-check confirmed that the pipeline produces accurate and culturally reasonable data.
> > >
> > > > **Q8: Comments**
> > > > **Q8-1:** In Figure 4, you should specify in the caption what performance (%) indicates. You should also specify what the readers see in a, b, c, d.
> > > > **Q8-3:** The authors state that the dataset “will be governed through protocols and licenses” but do not specify which protocols or licenses (e.g., CC-BY, CC-BY-NC-SA, or data-use agreements). Providing this information would clarify how the community can legally and ethically use CultureVerse and ensure compliance with open-science standards.
> > >
> > > We appreciate these detailed suggestions. Regarding Figure 4, we have updated the caption to explicitly define the "Performance (%)" metric and describe the specific contents of sub-figures (a), (b), (c), and (d). Please refer to the revised manuscript for these changes.
> > > Regarding the licensing, we intend to release the dataset under the CC-BY-NC-SA 4.0 license to ensure non-commercial research use. For some image data specifically, we will also include a custom Terms of Use and a hybrid protocol to respect original copyrights, which will be provided alongside the dataset release.
> > >
> > > Thank you again for your highly professional and insightful comments—they are invaluable not only for improving our work, but also for advancing the broader discourse on cultural understanding in the community. We hope our responses help clarify our contributions and foster a more positive assessment, and we warmly welcome any further discussion.

---

> ### Comment · Reviewer_pyUY · 2025-11-27
> **Acknowledgment of Author Response**
>
> Thank you for your detailed responses. After careful consideration, I have adjusted my soundness score accordingly.

---

### Official Review · Reviewer_AuDx · 2025-11-01

**Soundness:** 2
**Presentation:** 3
**Contribution:** 2
**Rating:** 4
**Confidence:** 4

**Summary:**

The authors develop a scalable data collection pipeline to collect CultureVerse, a multimodal dataset of culturally diverse images from 188 countries with English language questions (image recognition, cultural knowledge and scene understanding) covering 15 cultural topics.  The scale of the resulting corpus supports both model fine-tuning (producing CultureVLMs) and benchmarking (through a test set where the automatically generated questions were verified by humans).  They find that in general, VLMs exhibit regional disparities in cultural understanding, performing better on the Americas and less well for Asia and Africa.  This gap is narrowed through fine-tuning on CultureVerse.

**Strengths:**

The authors explore an urgent need in VLM development: multicultural awareness.

Their automated cultural knowledge extraction pipeline is able to handle more countries than existing datasets.

The authors are fairly robust in their evaluation – in particular, I liked their inter-culture / inter-region correlation experiments and their experiments that validate adaptation of their dataset to existing cultural understanding benchmarks like CVQA.

**Weaknesses:**

### Reliance on Wikipedia to source cultural concept examples

CultureVerse is extremely dependent on Wikipedia (in particular, English-language Wikipedia) for identification of country specific examples of its chosen cultural concepts.  While understandable, this means that it inherits the same limitations that Wikipedia exhibits by construction.  Of these limitations, the most critical are Wikipedia’s cultural biases which have been documented in work like https://arxiv.org/pdf/2305.14456.  Can the authors speak to this at all?

### Reliance on GPT4o instead of native representatives of each country / culture

While I am sympathetic to the need for scalable methods for automatically constructing a training dataset, I am nervous about the tremendous reliance on GPT4o in the construction of CultureVerse’s test set.  GPT4o is used to 1) filter out cultural concepts examples that lack distinct regional specificity, 2) generate comprehensive descriptions of country-specific cultural concept examples with information like location, characteristics, history and cultural significance and 3) formulate questions based on this GPT4o-generated cultural concept information.

The authors cite Li et al 2024c as arguing that GPT4o shows strong performance for text-based cultural understanding but 1) I am not sure that paper actually shows that -- could the authors cite relevant lines? and 2) by construction it would seem this benchmark has an implied ceiling of GPT4o’s cultural awareness which has been shown to be lacking in other work (e.g. ALM-Bench, CVPR 2025).  Can the authors speak to this at all?

Moreover, though the authors perform a manual review of their test set, this review is not conducted by native representatives of each country / culture.  Would these reviewers actually be able to catch errors in cultural knowledge introduced by GPT4o?  In several places (line 77, 163, 240, 244, 299-300) the authors refer to these reviewers as “expert” but it would seem they are not country / culture experts so much as they are trained on their annotation pipeline.  I would strongly encourage the authors to use a different term to describe them as currently the paper gives the impression that they are culture experts

As such, it would seem that CultureVerse is best suited as a fine-tuning corpus for evaluating VLMs on other cultural understanding benchmarks (as they do in their results on CVQA / CulturalVQA) rather than as a benchmark on its own.  Otherwise the community runs the risk of perpetuating the biases of GPT4o if it becomes a widely targeted leaderboard.

### Novelty

Relatedly, the past few years have seen the introduction of several new benchmarks for evaluating the cultural understanding of vision-language models.  Many of these benchmarks, like ALM-bench (CVPR 2025), are multilingual and involve native representatives of their selected cultures in their construction.  While CultureVerse is larger in scale, quality seems more important than quantity when discussing benchmarks.  Can the authors motivate the marginal value of CultureVerse compared to these other benchmarks in a more quantitative fashion?  Assuming the GPT4o-related concerns with CultureVerse described above have been addressed, it would be interesting to see model performance correlation on CultureVerse versus existing benchmarks like ALM-bench – are the benchmarks testing for different kinds of cultural understanding?

### Nitpicks / Typos

- line 131: “face more severe situation”
- line 157: “particularly FOR underrepresented groups”
- line 303: “regions with more countries yield larger datasets” – there are more countries in Africa than any other continent yet this region is among the smallest; as this is likely due to the limitations of Wikipedia (see above), it would be good to say so here as the rationale rather than merely country counts
- line 307: “multimodel”
- line 338: “pose challenges for comparable VLMs”
- line 341: “relies heavily on diverse and relevant image data” – can you add a citation for this?  perhaps https://proceedings.neurips.cc/paper_files/paper/2024/hash/c07d71ff0bc042e4b9acd626a79597fa-Abstract-Conference.html or some other
- line 370: “cultural knowledge often resides in a model’s memory, an aspect overlooked in VLM development” – can you add a citation for this?  perhaps https://arxiv.org/abs/2406.11665 or some other

**Questions:**

I found it curious that the authors chose to reserve the first image of each country-specific cultural concept for the test set (in particular, for the more common cultural concepts).  Does that mean that examples of every country’s specific cultural concepts (like sari) in its test set are also seen during training?  Why not hold out some concepts to test generalization?  If there’s examples of every class type in the training dataset, the benchmark becomes more of an image label learning benchmark than a cultural understanding benchmark.

To what degree is GPT-4o favored in this evaluation due to its core role in the construction of this corpus?  Would it be possible to re-run the pipeline for a subset of the test set and use Gemini instead of GPT-4o to measure this effect?

In Figure 5a, it would be nice to see the scores pre fine-tuning right now as it’s hard to make sense of the transferability without that baseline to compare against.

Some missing references:
- From Local Concepts to Universals: Evaluating the Multicultural Understanding of Vision-Language Models (EMNLP 2024)
- Crossmodal-3600: A Massively Multilingual Multimodal Evaluation Dataset (EMNLP 2022)

---

> ### Author Response · Authors · 2025-11-23
> **Response to Reviewer AuDx (1)**
>
> Thank you for your thoughtful review and for recognizing the urgency of multicultural VLM development, the scalability of our pipeline, and the rigor of our evaluation, including the inter-region analysis and cross-benchmark validation. We appreciate your constructive feedback and address all concerns below to improve our work further.
>
> > **Q1:** Reliance on Wikipedia to source cultural concept examples. CultureVerse is extremely dependent on Wikipedia (in particular, English-language Wikipedia) for the identification of country-specific examples of its chosen cultural concepts. While understandable, this means that it inherits the same limitations that Wikipedia exhibits by construction. Of these limitations, the most critical are Wikipedia’s cultural biases, which have been documented in work like https://arxiv.org/pdf/2305.14456. Can the authors speak to this at all?
>
> We fully agree that Wikipedia exhibits inherent cultural biases, such as the overrepresentation of Western topics and shallower descriptions for non-Western cultures. These biases often manifest in subjective narratives, political perspectives, or manifest downstream—for instance, in models trained on Wikipedia-derived n‑gram statistics, which exhibit a tendency to predict tokens reflecting Western-centric perspectives, as noted in the cited work.
>
> However, our methodology utilizes Wikipedia primarily to extract **visual concept entities** and their associated **objective factual information**, rather than generating open-ended subjective text. Consequently, the impact of these biases on our specific task is significantly minimized.
> Furthermore, for constructing a structured multimodal (image-text pair) corpus, **Wikipedia remains one of the few systematic, globally covered, and relatively clean data sources available**. While the cited paper suggests alternative "less Western" corpora like Twitter, such sources are often unstructured, noisy, low in information density, and prone to toxic content, making them unsuitable for building a high-quality, large-scale resource. Therefore, relying on Wikipedia represents a necessary and pragmatic trade-off to ensure data quality and scale, and the entire construction pipeline can be adapted to other related corpora.
>
> > **Q2:** Reliance on GPT4o instead of native representatives of each country/culture.
> > **Q5:** To what degree is GPT-4o favored in this evaluation due to its core role in the construction of this corpus?
>
> We thank the reviewer for this valuable and insightful discussion. Your observation that CultureVerse is best suited as a fine-tuning corpus is spot on, and this aligns with our paper's emphasis (where we primarily describe it as a 'corpus' or 'dataset' rather than a 'benchmark'). Indeed, the original goal of CultureVLM was **the scalable construction of a large-scale multimodal cultural corpus** to enhance model capabilities. We only carved out a portion of the data as a test set because, during development, we found that existing community benchmarks were either unavailable (e.g., CVQA/CulturalVQA had just been released and did not make test answers public) or had limited regional coverage. We extended our efforts to construct this benchmark primarily to monitor/analyze global cultural understanding and facilitate model iteration.
>
> Regarding the annotation process, we believe our pipeline ensures high quality and validity, comparable to methods that claim to rely solely on locals. For LLM-based filtering, extensive experiments on benchmarks (e.g., CVQA and ALM-Bench) have demonstrated that GPT-4o is currently one of the top-performing models in multicultural understanding. For human annotation, culture is a highly granular, region-specific concept. Even within the same country/region, individuals from different cities/sub-regions may not be fully familiar with each other's local customs. By training our annotators for a month and requiring cross-verification through web searches (ensuring multiple independent sources confirm the information), we were able to achieve a level of accuracy comparable to that of knowledgeable local annotators.

---

> ### Author Response · Authors · 2025-11-23
> **Response to Reviewer AuDx (2)**
>
> > **Q3:** Novelty. Relatedly, the past few years have seen the introduction of several new benchmarks for evaluating the cultural understanding of vision-language models. Many of these benchmarks, like ALM-bench (CVPR 2025), are multilingual and involve native representatives of their selected cultures in their construction. While CultureVerse is larger in scale, quality seems more important than quantity when discussing benchmarks. Can the authors motivate the marginal value of CultureVerse compared to these other benchmarks in a more quantitative fashion? Assuming the GPT4o-related concerns with CultureVerse described above have been addressed, it would be interesting to see model performance correlation on CultureVerse versus existing benchmarks like ALM-bench – are the benchmarks testing for different kinds of cultural understanding?
>
> We appreciate the contemporaneous contributions from the community, such as ALM-Bench, which signal the growing importance of cultural understanding in LLMs/VLMs. Regarding the comparison, we note that ALM-Bench relies on GPT-4o to fully synthesize QA pairs and translate them, followed by native speaker revision. This pipeline may still inadvertently retain some of GPT-4o’s intrinsic biases. In contrast, our approach uses GPT-4o primarily for **extracting** cultural knowledge from the Wiki Document, rather than relying on the model to hallucinate or synthesize internal knowledge for QA generation.
>
> While we acknowledge that native speaker curation can enhance the quality of benchmark subsets, we believe CultureVerse also achieves a superior standard as a large-scale corpus. In terms of performance correlation, both benchmarks exhibit very similar trends: GPT-4o and Gemini-1.5-Pro consistently outperform other models on both CultureVerse and ALM-Bench. This alignment reinforces the validity and high quality of our benchmark subset as a reliable evaluation tool.
>
> Finally, we emphasize that while all LLM-assisted datasets inevitably contain some degree of bias, our corpus offers distinct advantages: it is more comprehensive, scalable, and specifically designed to enhance cultural training. Furthermore, our manually annotated test subset also ensures high-quality evaluation standards.
>
> > **Q4:** I found it curious that the authors chose to reserve the first image of each country-specific cultural concept for the test set (in particular, for the more common cultural concepts). Does that mean that examples of every country’s specific cultural concepts (like sari) in its test set are also seen during training? Why not hold out some concepts to test generalization? If there are examples of every class type in the training dataset, the benchmark becomes more of an image label learning benchmark than a cultural understanding benchmark.
>
> We view the improvement of cultural understanding primarily as a **knowledge-intensive task**. Our goal is for the model to acquire the most comprehensive cultural knowledge possible, synthesizing information from all available data. Therefore, we maximized the utilization of collected data during training. However, we fully recognize the importance of evaluating generalization. To address this, we assess the model's generalization capabilities through cross-region evaluations and performance on external cultural benchmarks. These experiment results demonstrate that the model is not merely memorizing image labels but is learning transferable cultural representations.
>
> > **Q5:** In Figure 5a, it would be nice to see the scores pre fine-tuning right now, as it’s hard to make sense of the transferability without that baseline to compare against.
>
> We appreciate the suggestions regarding Figure 5a. We have revised the paper to include the pre-fine-tuning baselines in Figure 5a for better comparison. Please check it in the revised manuscript.
>
> > **Q6:** Typos and Missing References
>
> We sincerely appreciate your detailed feedback. Such a thorough review is rare and highly valued, given the current volume of submissions. In response, we have carefully corrected all typographical errors throughout the manuscript and carefully read all the reference papers you mentioned, gaining significant insights that have further deepened our understanding of cultural nuances. **Please check the portions in blue font in the revised version**.
>
> We sincerely hope our clarifications and revisions help strengthen your perception of our work’s contribution, and we would be delighted to address any further questions or continue the discussion at your convenience. Thank you again for all your suggestions to improve our work!

---

### Official Review · Reviewer_sbZ9 · 2025-11-01

**Soundness:** 2
**Presentation:** 3
**Contribution:** 3
**Rating:** 4
**Confidence:** 3

**Summary:**

This paper introduces CultureVLM, a framework for characterizing and improving cultural understanding in Vision-Language Models (VLMs). The authors develop CultureVerse, a large-scale multimodal cultural dataset covering 188 countries/regions, 15 cultural topics, and 228K samples across three question types: image recognition, cultural knowledge, and scene understanding. The paper evaluates 16 VLMs and demonstrates significant regional disparities in cultural understanding, with models performing best on Western cultures (Americas, Europe) and worst on Asian and African cultures. The authors fine-tune several models on CultureVerse, showing improvements in cultural understanding while maintaining general capabilities.

**Strengths:**

- **Timely and Socially Significant Problem Formulation**
The work addresses a pressing gap in multimodal AI - the lack of cultural inclusivity in models. As prior studies highlight, most LLMs and VLMs align strongly with Western cultural norms.

- **Comprehensive Dataset Construction Framework**
The CultureVerse pipeline is clearly described and methodologically coherent. The combination of automated Wikipedia scraping followed by concept refinement and multi-tiered QA ensures scalability while maintaining reasonable data quality.

-  **Thorough Experimental Analysis**
The paper provides extensive analysis across multiple dimensions - regional disparities, task characteristics, model scaling effects, and generalization capabilities. The finding that cultural understanding doesn't scale linearly with model size (e.g., Llama 3.2-11B matching Qwen 2-72B performance) is particularly noteworthy.

**Weaknesses:**

- **Methodological Concerns with Dataset Construction**
The heavy reliance on GPT-4o for concept extraction and question generation raises concerns about systematic biases being embedded in the dataset. While the authors claim GPT-4o has "strong performance for text-based cultural understanding," this assertion lacks adequate validation. Recent work has shown that even advanced LLMs exhibit cultural biases , potentially propagating these biases into the benchmark.​​

- **Evaluation Methodology Limitations**
The exclusive use of multiple-choice questions limits the assessment of nuanced cultural understanding. The authors acknowledge this limitation but don't adequately address how this format may not capture the complexity of cultural competence. Some more fine-grained evaluations would strengthen the work.

- **Insufficient Analysis of Data Quality**
 While the authors report high accuracy rates for automated annotations (93-99%), the validation is based on only 10 human annotators with limited cultural expertise. The paper lacks detailed analysis of inter-annotator agreement or validation across diverse cultural perspectives. This is particularly concerning given the global scope of the dataset.

- **Lack of fine-grained analysis**
The finding about regional disparities, while important, is not surprising given known biases in training data. The paper would benefit from deeper analysis of why these disparities exist and how they can be systematically addressed.​

**Questions:**

Please refer to the Weaknesses section.

---

> ### Author Response · Authors · 2025-11-23
> **Response to Reviewer sbZ9**
>
> We sincerely appreciate your recognition of the significance of our problem formulation, the robustness of the CultureVerse construction pipeline, and the depth of our experimental analyses. We are also grateful for your constructive feedback, which has helped us further strengthen the clarity and rigor of our work. Below, we provide detailed responses to address your concerns.
>
> > **Q1:** Methodological Concerns with Dataset Construction.
>
> We understand your concern regarding potential biases and validation of GPT-4o on cultural understanding. Extensive experiments on benchmarks such as CVQA and ALM-Bench **have demonstrated that GPT-4o is currently one of the top-performing models regarding multicultural understanding**, exhibiting minimal cultural bias. Furthermore, our methodology **restricts GPT-4o’s role to extracting structured data** directly from Wikipedia documents. By leveraging the model primarily for its linguistic processing capabilities to parse external sources—rather than relying on its internal parametric knowledge, we significantly mitigate the influence of the model’s intrinsic biases. Finally, the resulting benchmark subset underwent rigorous human verification to ensure accuracy and prevent the propagation of any residual bias.
>
> > **Q2:** Evaluation Methodology Limitations. The exclusive use of multiple-choice questions limits the assessment of nuanced cultural understanding. The authors acknowledge this limitation but don't adequately address how this format may not capture the complexity of cultural competence. Some more fine-grained evaluations would strengthen the work.
>
> **Multiple-choice questions remain the standard evaluation format in both concurrent cultural datasets (e.g., CVQA) and general benchmarks (e.g., MMLU)**. We adopted this format because it ensures objective and unbiased scoring, avoiding the high costs and potential variability associated with LLM-as-a-Judge or manual evaluation. We acknowledge that more fine-grained evaluation methods exist, such as interactive assessments conducted by local native speakers. However, given that the primary focus of this paper is on the scalable construction of the dataset/corpus itself, we have prioritized scalability and reproducibility. We plan to address these more resource-intensive, human-centric evaluation methods in future work.
>
> > **Q3:** Insufficient Analysis of Data Quality.
>
> Thank you for the insightful comment. The annotation of cultural data is a interesting topic with many details worth discussing, which we realized through both self-annotation and working with hired annotators. When annotating culture from our own nations, we discovered that due to differences between cities or sub-regions, it is often difficult to deeply understand the culture of other parts of the same country. Ultimately, we still needed to use search engines to confirm facts. Thus, being a native speaker does not necessarily imply a deep understanding of every specific cultural detail. We hired 10 people, a small-scale team, to annotate the 10k concepts in the test subset over the course of a month. Their instruction was to verify every piece of data by consulting multiple sources via search engines. We believe that a small team working over a long period is easier to manage and better guarantees the consistency and quality of the test set. Relevant details have also been described in the Appendix D. There was over 80% consistency among annotators. After the annotation pipeline, we also performed random quality inspections on the test set, finding that the error rate remained below 1%.
>
> > **Q4:** Lack of fine-grained analysis.
>
> Thank you for the feedback. While we acknowledge that cultural bias is an inherent challenge in LLMs and VLMs, we believe a multi-faceted analysis remains crucial to understanding its extent and mechanics. Our paper attempts to **go beyond the general observation of bias by exploring specific questions**, e.g., Do all models exhibit similar regional disparities? What is the precise magnitude of these differences? How does cultural understanding scale with model size? Which specific types of cultural concepts are most difficult to recognize? How does the model perform after training—can it generalize across regions? By addressing these questions, our analysis also aims to **provide actionable insights for future systematic improvements**, such as the need to intervene at the pre-training data level, target data collection for low-resource regions, and the importance of providing detailed conceptual information during training rather than relying solely on simple image-text pairs.
>
> Thank you again for the insightful feedback. We hope these clarifications could enhance your understanding and evaluation of our work.

---

### Note · Authors · 2026-01-06

I have read and agree with the venue's withdrawal policy on behalf of myself and my co-authors.